# Weak sensitivity of the terrestrial water budget to global soil texture maps in the ORCHIDEE land surface model

Salma TAFASCA[1], Agnès DUCHARNE[1], Christian VALENTIN[2]

[1] METIS (Milieux Environnementaux, Transferts et Interactions dans les Hydrosystèmes et les Sols), Institut Pierre Simon Laplace (IPSL), Sorbonne Université, CNRS, EPHE, Paris, France

[2] iEES-Paris (Institut d'Ecologie et des Sciences de l'Environnement de Paris), Sorbonne Université, CNRS, INRA, IRD, Paris, France

*Correspondence to*: Salma Tafasca (salma.tafasca@upmc.fr)

**Abstract**

Soil physical properties play an important role for estimating soil water and energy fluxes. Many hydrological and land surface models (LSMs) use soil texture maps to infer these properties. Here, we investigate the impact of soil texture on soil water fluxes and storage at different scales using the ORCHIDEE LSM, forced by several complex or globally-uniform soil texture maps. At point scale, the model shows a realistic sensitivity of runoff processes and soil moisture to soil texture, and reveals that loamy textures give the highest evapotranspiration and lowest total runoff rates. The three tested complex soil texture maps result in similar water budgets at  all scales, compared to the uncertainties of observation-based products and meteorological forcing datasets, although important differences can be found at the regional scale, particularly in areas where the different maps disagree on the prevalence of clay soils. The three tested soil texture maps are also found to be similar by construction, with a shared prevalence of loamy textures, and have a spatial overlap over 40% between each pair of maps, which explains the overall weak impact of soil texture map change.   A useful outcome is that the choice of the input soil texture map is not crucial for large-scale modelling but the added-value of more detailed soil information (horizontal and vertical resolution, soil composition) deserves further studies.

## 1. Introduction

Land surface models (LSMs) simulate water and energy fluxes at the interface between the land surface and the atmosphere. They were developed for continental to global scales to provide realistic land boundary conditions to climate models (Remaud et al., 2018), and to investigate the water, energy and carbon cycles at the Earth surface, and the related natural resources and risks (Guimberteau et al., 2017; Haddeland et al., 2011; Sterling et al., 2013; Zhao et al., 2017). By lack of sufficient spatial coverage for detailed soil properties, LSMs, like many physically-based hydrological models, rely on pedotransfer functions (PTF), which relate available soil information to the required soil properties (Looy et al., 2017; De Lannoy et al., 2014). The simplest approach, still used by most LSMs, relies on soil texture, as classified by the US Department of Agriculture (USDA) into 12 soil classes based on the percent of sand, silt and clay particles (USDA Soil Survey Staff et al., 1951). Look-up tables relate these broad texture classes to multiple soil properties, usually with one single central value for each class and property,

as found in Cosby et al. (1984) and Carsel and Parrish (1988) for the Clapp and Hornberger (1978) and Van Genuchten (1980) soil water models, respectively. In this framework, several global soil texture maps are used by LSMs, with different resolutions and soil texture distributions: based on the 1:5,000,000 FAO/UNESCO Soil Map of the World (FAO/UNESCO, 1971-1981), itself based on soil surveys defining 106 soil units, Zobler (1986) and Reynolds et al. (2000) provided soil texture maps at a resolution of 1° and 5 arc-min respectively, for depths of 30 and 100 cm for Reynolds et al. (2000), and 30 cm for Zobler (1986); the FAO/UNESCO Soil Map of the World was updated as the Harmonized World Soil Database (HSWD), produced at 30 arc-sec by including new regional and national soil information (Nachtergaele et al., 2010; Batjes, 2016); the soil texture map of the 1-km SoilGrids database (Hengl et al., 2014), recently updated at 250 m (Hengl et al., 2017), is not independent from the above FAO/UNESCO global soil maps, but also relies on large number of national and international soil profile databases, combined with automated spatial prediction models. Both HSWD and SoilGrids soil texture maps are available at seven depths ranging from 0 cm to 2 m.

Most studies concluding that soil texture exerts an important impact on soil hydrology were conducted at small to medium scales, either through site measurements (e.g. An et al., 2018; Song et al., 2010), or regional-scale and multi-site data analysis (Lehmann et al., 2018; Wang et al., 2009) and model sensitivity analyses. Using a mesoscale hydrologic model over the Mississippi river basin, Livneh et al. (2015) compared two different soil texture maps, and the more spatially detailed one better reproduced hydrologic variability and extreme events. With the Noah LSM over China, Zheng and Yang (2016) found that the sensitivity of the simulated water budget to soil texture was dependent on climate, soil moisture being less sensitive to soil texture in arid areas, while evapotranspiration and runoff showed the highest sensitivity in the transitional zones. Li et al. (2018) confirmed these results over the Tibetan Plateau but showed additional influence of the vegetation cover on the sensitivity to soil texture, as also found over the US (Xia et al., 2015). At a global scale, De Lannoy et al. (2014) developed an improved soil texture map for the Catchment LSM, by merging several texture and organic material maps. Combined with updated PTF, this new map offered modest yet significant improvements of the simulated hydrology compared to various point-scale measurements. Related studies revealed a strong impact of soil water-holding capacity and its spatial patterns using the first generations of LSMs, but with bucket-type soil hydrology instead of Richards equation (Milly & Dunne 1994; Ducharne & Laval, 2000).

Here, we aim at exploring more systematically the impact of soil texture on the water budget from point to global scale, using a state-of-the-art LSM with physically-based soil hydrology, and multiple input soil texture maps. After presenting the model and soil texture maps used in this work, the results are presented as follows. We first provide an analysis of the similarities and differences between the different soil maps, then, we evaluate the point-scale response of the model to different soil textures to make sure it displays a reliable behaviour. This point-scale response is then analysed from a geographic point of view, with a comparison to a distributed observation-based evapotranspiration product, and a focus is made on areas with a large sensitivity to the soil texture maps. We finally explore how the magnitude and significance of the simulated

evapotranspiration response changes with the scale of analysis up to the land scale, defining the terrestrial water budget. The closing section summarizes the main conclusions of the study, and discusses its limitations and perspectives.

## 2. Materials and Methods

### 2.1. Soil texture in the ORCHIDEE LSM

ORCHIDEE (ORganizing Carbon and Hydrology in Dynamic EcosystEms) is the land component of the IPSL (Institut Pierre-Simon Laplace) climate model, and describes the complex links between vegetation phenology and the water, energy and carbon exchanges at the land surface (Krinner et al., 2005). We use here the version of ORCHIDEE developed for CMIP6 (Eyring et al., 2016) and detailed in forthcoming papers (Boucher et al., 2019; Cheruy et al., 2019; Ducharne et al., in prep).

The physically-based soil hydrology scheme solves the vertical soil moisture redistribution based on a multi-layer solution of the saturation-based Richards equation, using a 2-m soil discretized into 11 soil layers of increasing thickness with depth (de Rosnay et al., 2002). Infiltration is processed before soil moisture redistribution, owing to a time-splitting procedure inspired by the model of Green and Ampt (1911), with a sharp wetting front propagating like a piston (d'Orgeval et al., 2008; Vereecken et al., 2019). The unsaturated values of hydraulic conductivity and diffusivity are given by the model of Mualem (1976) - Van Genuchten (1980).

In each grid cell, the corresponding parameters (saturated hydraulic conductivity $K_s$, inverse of air entry suction $\alpha$, shape parameter $m$, porosity, and residual moisture) are taken from Carsel and Parrish (1988), as a function of the dominant USDA soil texture class, itself derived from an input soil texture map. The tabulated values of the different soil parameters are displayed in Figure S1 for each USDA class. Soil texture is assumed to be uniform over the soil column in ORCHIDEE, which does not permit to distinguish several soil horizons. However, $K_s$ decreases exponentially with depth, to account for the effects of soil compaction and bioturbation, as introduced by d'Orgeval et al. (2008) following Beven & Kirkby (1979). It must also be noted that the horizontal variations of $K_s$ are taken into account by an exponential probability distribution, but only for calculating infiltration and surface runoff (Entekhabi & Eagleson, 1989; Vereecken et al., 2019). The soil texture also influences heat capacity and conductivity, and heat diffusion is calculated with the same vertical discretization as water diffusion in the top 2m, but extended to 10 m (Wang et al., 2016). Evapotranspiration is described by a classical bulk aerodynamic approach, distinguishing four sub-fluxes: sublimation, interception loss, soil evaporation, and transpiration. The latter two are directly coupled to soil water redistribution, and depend on soil moisture and properties, which control how the corresponding rates are reduced compared to the potential rate: transpiration is limited by a stomatal resistance, increasing when soil moisture drops from field capacity to wilting point (which both depend on soil texture as detailed in Supplementary S1); soil evaporation is not limited by a resistance, but only by upward capillary fluxes, which control the soil propensity to meet the evaporation demand (d'Orgeval et al., 2008; Campoy et al., 2013) . Evapotranspiration also depends on the vegetation of each grid-cell, described here as a mosaic of up to 15 plant functional types (PFTs), based on the global land cover map

used in the IPSL simulations for CMIP6 (Boucher et al., 2019). In each PFT, root density is assumed to decrease exponentially with depth, with a PFT-dependent decay factor. The resulting root density profile is combined to the soil moisture profile and a water stress function depending of field capacity and wilting point to define the integrated water stress factor of each PFT on transpiration.

This flux is also coupled to photosynthesis, which depends on soil moisture, light availability, $CO_2$ concentration, and air temperature, following Farquhar et al. (1980) and Collatz et al. (1992) for C3 and C4 plants, respectively. The resulting carbon assimilation is allocated to several vegetation pools, including leaf mass thus leaf area index (LAI), owing to a dynamic phenology module called STOMATE (Krinner et al., 2005). It must be underlined that LAI has an important influence on the partition between soil evaporation and transpiration, via the fraction that is effectively covered by foliage, which increases exponentially with LAI with a coefficient of 0.5, also controlling light extinction through the canopy (Krinner et al. 2005). This fraction contributes to transpiration and interception loss, while the complementary fraction is assumed to be bare of vegetation, and only contributes to soil evaporation.

### 2.2. Simulation protocol

We performed nine global-scale simulations with ORCHIDEE (tag 2.0), using different soil texture maps and climatic forcing datasets (Table 1). The analysed period is 1980-2010, following a 20-year warm-up since 1960 to provide accurate initial conditions. Atmospheric forcing datasets being known to exert a first-order influence on LSM results (Guo et al., 2006; Yin et al., 2018), we used two different datasets to drive our simulations, to compare the related uncertainties to the ones coming from the different soil texture maps. Both datasets were constructed at a 0.5° resolution by downscaling and bias-correcting an atmospheric reanalysis. All simulations but one use the GSWP3-v1 meteorological dataset (Kim, 2017), with a 3-hourly time step, and based on the 20[th] Century Reanalysis (20CR; Compo et al., 2011). In contrast, simulation EXP1 uses the 6-hourly CRU-NCEP-v7 meteorological dataset (Wei et al., 2014), based on the NCEP/NCAR reanalysis (Kalnay et al., 1996), and extended beyond 1957-1996 in near real-time. Both meteorological datasets were selected for the off-line CMIP6 simulations (van den Hurk et al., 2016).

The three simulations EXP2 to EXP4 rely on complex soil texture maps to define the dominant texture class of each 0.5° grid cell (Figure 1): the 1° map of Zobler (1986) originally contains 5 soil textural classes, but is simplified by ORCHIDEE into three USDA texture classes (Sandy Loam, Loam, and Clay Loam); the 5-arc-min map of Reynolds et al. (2000) uses the USDA classification and we used directly the 30-cm map; the third map was upscaled from the 1km SoilGrids map at the 0 cm depth (Hengl et al., 2014) by selecting the dominant soil texture in every 0.5° pixel. This map was provided at a 0.5° resolution by the Soil Parameter Model Intercomparison Project (SP-MIP, Gudmundsson & Cuntz, 2017), which aims at quantifying to which degree the differences between LSMs result from soil parameter specification, and will thus be referred to as the SP-MIP map in the following.

In addition, we tested four spatially uniform texture maps, corresponding to the Loam, Loamy Sand, Silt, and Clay classes (EXP6 to EXP9), to analyse the importance of the spatial variability of soil texture on the terrestrial water budget. These simulations were defined by SP-MIP, and rely on hydraulic parameter values given by Schaap et al. (2001) for each USDA class. We ran an additional simulation (EXP 5) with the SP-MIP map and the soil parameters of Schaap et al. (2001) to quantify the difference induced by this PTF compared to the default PTF of ORCHIDEE (Carsel & Parrish, 1988) used with the SP-MIP map in EXP4. It must be noted that the five simulations based on the soil parameters of Schaap et al. (2001) also differ from the four others (EXP1 to EXP4) because the decrease of $K_s$ with depth is relaxed, to comply with the SP-MIP protocol.

### 2.3. Calculation of median diameter *dm* for each of the 12 USDA soil texture classes

Every texture class is represented by a polygon in the USDA textural triangle (Fig. 1d). For each texture class, we located the centroid of the corresponding polygon to obtain a central value of the composition in clay, silt and sand particles (Table 2). These clay, silt and sand particles have various diameters, respectively ranging in [0, 2μm], [2μm, 50μm] and [50μm, 2000μm] (USDA; Staff, 1951). To construct the particle-size distribution curve of each texture class (Fig. 2), we further assumed that clay, silt and sand particle diameters are uniformly distributed in the latter intervals. The median diameter of each texture class is then obtained by intersecting the corresponding curve with a cumulative value of 50%, such that half of soil particles reside above this point, and half reside below this point. The resulting median diameters are listed in Table 2. Carsel and Parrish (1988) provide the mean content of sand, silt and clay for each soil texture, but their estimations are based on American soil surveys, which might not be representative of the whole globe, so we preferred to use the composition of the of the polygon centroids. Note that using the mean composition of by Carsel and Parrish (1988) leads to very similar results.

### 2.4. Evaluation datasets

To assess the realism of our simulations, we use three different datasets. Jung et al. (2010) constructed a series of global 1° evapotranspiration maps at the monthly time step from 1982 to 2008, by interpolating in situ eddy-covariance measurements from the FLUXNET network using machine learning algorithms and ancillary geospatial information (land surface remote sensing and meteorology). GLEAM (Martens et al., 2017) is another series of global evapotranspiration maps, provided by at the 0.25° resolution and the daily time step over 1980-2015. They strongly rely on remote-sensing datasets (radiation, precipitation, temperatures, surface soil moisture, vegetation optical depth, snow water equivalents), used as input to an evapotranspiration model based on Priestley and Taylor (1972). Finally, Rodell et al. (2015) quantified the mean annual fluxes of the water cycle at the beginning of the 21[st] century, at a coarser scale (continents and majors ocean basins) but with the aim of providing consistent estimates of precipitation, evaporation, and runoff, by combining in situ and satellite measurements, data assimilation systems, and multiple energy and water budget closure constraints.

## 3. Results

### 3.1. Comparison of the tested soil texture maps

Whichever the complex soil texture map, Loam is by far the most dominant texture (Table 1), covering between 44 to 64 % of the land surface. Other important soil textures in all maps are Sandy Loam, Clay Loam and Sandy Clay Loam, and these four medium textures alone cover 81, 86 and 100 % of land based on Reynolds, SP-MIP, and the simplified Zobler map, respectively. The Silty Clay, Silty Clay Loam and Sandy Clay classes are poorly present in all three maps: altogether, they cover 0.9, 0.2 and 0 % of land based on SP-MIP, Reynolds and the simplified Zobler map, respectively. The Silt texture class is absent from Reynolds and Zobler maps, while it found in the SP-MIP map, but only to fill the no-data land points (3.3%). To better document the differences and similarities between the three soil texture maps, we also quantified the spatial overlap between each pair of complex maps (Table 3). It is always more than 41%, and the best agreement is found between the Reynolds and Zobler maps (52%). Nonetheless, this leaves 48 % of the grid cells (in the best case) where the soil texture does change.

To explore if it changes for similar or very different soil textures, we compared several groups of three maps derived from the tested soil texture maps: the maps of the corresponding particle diameter $dm$ (section 2.*3*), of $K_s$ (as provided by the PTF, thus not including the impact of roots nor soil compaction), soil porosity $\theta_s$, field capacity $\theta_{fc}$, wilting point $\theta_w$, and available water content (AWC, integrated over the 2-m soil column). The values of these soil parameters for the 12 USDA soil texture classes are detailed in Supplementary S1 and depicted in Figure S1. Table 4 shows that, whichever the soil parameter, the difference of spatial mean between the three maps is smaller than the mean spatial standard deviation, even for the least variable map (Zobler). This demonstrates the large similarity of the three complex soil maps tested in our paper, not only regarding soil texture itself (as summarized by $dm$), but also, very logically, for the derived soil hydraulic parameters. This similarity is also confirmed by the spatial correlations between each pair of maps, always positive, the best correlations being found between Zobler and Reynolds for most parameters (always larger than 0.35, and up to 0.58), and the weakest between the SP-MIP map and Reynolds.

### 3.2. Point scale sensitivity to the 12 USDA texture classes

To check if the ORCHIDEE model displays a realistic response to soil texture, we examined how the pluri-annual means of the main water budget variables relate to soil texture (Fig. 3). We clustered all the points with a similar texture, and sorted the texture classes based on their median particles diameter (section2.3). The mean fluxes were also divided by mean precipitation to reduce the effect of misleading texture-climate associations, as between sandy classes and arid climates. We focused on EXP2, since the Reynolds map exhibits the largest range of soil textures (11 different classes).

The simulated total soil moisture (over the 2-m soil depth), drainage and surface runoff exhibit a clear monotonic response to soil texture (sorted by median diameter). Increasing soil moisture for finer textures is explained by their higher

water retention and field capacity. The opposite responses of drainage and surface runoff (Fig. 3f-g) both result from higher permeability in coarser soils, enhancing drainage and infiltration at the soil surface, thus reducing surface runoff. These responses to soil texture are coherent with experimental results (e.g. An et al., 2018; Song et al., 2010).

As it sums up two opposite responses, total runoff shows a larger spread and a non-monotonic (convex) behaviour, with smaller total runoff for medium textures. The opposite response (concave) is found for evapotranspiration (Fig. 3d), because precipitation is partitioned between evapotranspiration and total runoff in every grid cell. The highest evapotranspiration rates found for medium textures is consistent with the high available water capacity for these loamy textures (Fig. S1). Transpiration, however, increases as soil gets coarser (Fig. 3c), with two explanations probably acting together. Firstly, the increase of matric potential when the texture gets finer, as shown in Figure S1 for particular values of the potential, defining the wilting point, field capacity and air entry suction point ($1/\alpha$), makes root uptake thus transpiration more difficult for a given soil moisture if the soil texture is finer. Secondly, the high conductivity of coarse soils enhances water infiltration at the soil surface, quickly available for plant uptake. The increase of $K_s$ for coarse textures also explains the associated drainage increase when its dependence on mean precipitation is filtered (Fig. 3f). The fact that soil moisture decreases when drainage and transpiration get higher indicates that annual mean soil moisture is the result more than the cause of these fluxes.

Soil evaporation shows more variability within a soil texture class than between the different soil texture classes (Fig. 3b), showing this flux strongly depends on other factors, like temperature, leaf area index, etc. (Martens et al., 2017; Wang et al., 2018). To filter their spurious effects, we also analysed in Figure 4 the effect of changing soil texture  at the point-scale, thus under similar climatic and land cover conditions. Figure 4 shows the changes occurring where a soil texture class in the Reynolds map is replaced by another in the SP-MIP map. The Zobler map was excluded from this analysis since it contains only three soil texture classes. Switching maps from Reynolds to the SP-MIP map (i.e. from EXP2 to EXP4) results in a majority of land points with unchanged texture, and thus, identical simulated variables. These land points are represented by the diagonal pixels of the matrices and correspond to 41.2% of the land surface. Land points with coarser texture in the SP-MIP map represent 34.1% of the land surface (upper side of the diagonal line in the matrices) against 24.7% for finer textures (lower side of the diagonal line in the matrices).

Figure 4 highlights that simulated soil evaporation decreases from fine to coarse textures, so that capillary retention, which is the main limiting factor to soil evaporation in ORCHIDEE, depends more strongly on soil moisture (higher for fine soils; Fig. 4e) than on intrinsic capillary forces (stronger for fine soils). We fail to see this behaviour in Figure 3, which is likely due to the greater impact of diverse climatic conditions and vegetation associated with every soil texture. Figure 4 also confirms the results of Figure 3 for the other variables, including the decrease of soil moisture with coarser soils and the larger impact of soil texture on surface runoff and drainage than on transpiration and soil evaporation.  In particular, we find that replacing fine textures with coarse textures (above the first diagonal of the matrices) results in higher drainage (due to the

higher permeability of coarse-textured soils) and lower surface runoff, with changes that can exceed 1 mm/d in absolute value for some textural changes (all involving medium texture classes).

The convex behaviour of total runoff with soil texture can also be seen in Figure 4h, which is antisymmetric along the two diagonals, thus defining four different kinds of total runoff response to soil texture change. This behaviour results from the fact that total runoff sums up two variables with an opposite response to soil texture change (surface runoff and drainage), the net response depending on the dominant component. Hence, changes to medium textures from either coarse or fine textures (left and right red triangles in Fig. 4h) lead to reduced total runoff, owing to reduced surface runoff in the first case, and reduced drainage in the second. In contrast, changes from medium texture to either coarse or fine textures lead to increased runoff (bottom and top blue triangles in Fig. 4h), owing to increased surface runoff or drainage, respectively. This pattern thus means that the medium textures correspond to the smallest total runoff. By means of long-term water conservation, the opposite patterns are found for total evapotranspiration changes (Figure 4d), because of the opposite responses of soil evaporation and transpiration to soil texture, and supporting the concave response of this flux to soil texture found in Figure 3.

### 3.3. Spatial patterns of simulated fluxes and evapotranspiration bias

Although ORCHIDEE exhibits a clear and physically-based response to soil texture at point-scale, the use of three different realistic soil texture maps (EXP2, EXP3, and EXP4) results in rather similar spatial distributions of the simulated fluxes. We mostly focus on evapotranspiration (Fig. 5, Fig. S2), since comparison is possible with a spatially-distributed observation-based product (GLEAM). At a grid cell scale, changing the soil texture map (Fig. 5a-c) results in weak changes in simulated evapotranspiration, which are statistically significant over less than 35% of the land surface, against 77% when switching the climate forcing (Fig. 5d). The very weak changes in evapotranspiration maps when switching from a uniform to a complex soil texture map (Fig. 5a) show that the spatial variability of soil texture is a weak driver of the spatial variability of evapotranspiration. In agreement with the concave response of evapotranspiration to soil texture (section 3.2), the largest increases are found when switching from very coarse or very fine textures to medium ones. This explains the dominance of evapotranspiration increase in the example cases of Figures 5a-b, since the Zobler and uniform Loam maps have the largest areal fractions of Loam (Table 1).

Consistently, the evapotranspiration biases are overall similar whichever the soil texture map (Fig. 5e-g), while climate forcing uncertainty appears as a first order driving factor of the bias patterns (with visible differences between Figs. 5g and 5h). We find that the simulated evapotranspiration better matches GLEAM with CRU-NCEP in equatorial rain belts, and with GSWP3 in the mid-latitudes. In a few spots, however, the different soil maps induce large changes of evapotranspiration biases, especially in Central Africa, Central America, India and the Amazon basin, which are discussed in the following subsection. The other simulated hydrologic variables display a stronger sensitivity to soil texture maps, in agreement with section 3.2, but it remains weak and predominantly insignificant in front of inter-annual variability (Fig. 6, S3).

To provide a point-scale quantification of the differences between the three complex soil maps and the resulting simulated variables, we mapped the standard deviation of each group of three maps, using the mean diameter (*dm*) of each texture class to get a quantitative proxy in case of texture (Fig. 7). Although the quantitative meaning of standard deviation can be questioned when calculated from a sample of three values, we used it here as a simple metric of similarity/difference between the three complex maps, and to identify points/regions where the three maps are all consistent (small standard deviation), or where at least one of them is departing (high standard deviation). Compared to the standard deviation of log(*dm*), the ones of the simulated fluxes are weak (less than 10 % of the maximum value) over larger fractions of the globe. They are also smaller than the local annual mean values of the variable itself, as shown by comparison to Figure S2 for evapotranspiration (not shown for the other variables).

Areas which stand out with high standard deviations in all maps are the four regions noted above, where the standard deviation between the three texture maps is very important (Fig. 7a). Aside from these areas, the tropical humid zones (South-East Asia, Indonesia) show rather large standard deviations of surface runoff and drainage (Fig. 7d,f), but without large standard deviation of log(*dm*), so this is rather due to the high values of these fluxes in these very humid zones. The overall resemblance between the standard deviation maps of soil texture on the one hand, and the simulated hydrologic variables on the other hand can be quantified at global scale by a spatial correlation coefficient, ranging between 0.49 for transpiration to 0.79 for soil moisture. The latter variable is the most impacted by soil texture change, as supported by this large correlation coefficient, and the large standard deviations on Figure 7b.

### 3.4. Regional zooms on greatly impacted areas

Figure 8 displays the four 40° x 60° areas where the different soil maps can lead to strongly different evapotranspiration biases, with a strong link to the (mis)representation of Clay soils, since the largest changes in evapotranspiration and total runoff are expected where soil texture changes between medium (loamy) and extreme (Clay or Sand) textures (Figs. 3 and 4). The Sand soil texture, however, does not induce a large impact on the simulated hydrological fluxes, as it is mostly found in arid areas where water is a limiting factor. This is the case in the Arabian Peninsula (Fig. 8b) and the Sahara, where the sandy soils mapped in the SP-MIP map are absent in Zobler and only weakly present in Reynolds, but the evapotranspiration bias hardly changes and remains negative.

In Tropical South America and Central Africa (Fig. 8c,d), the Reynolds map shows a larger presence of Clay compared to the other two maps, part of which results in an important negative evapotranspiration bias. When compared to the FAO soil order map (Fig. S4), it is found that the Clay class of the Reynolds map gathers different soil orders, including (i) Vertisols, which consist of swelling clay (smectites) with low permeability, and mostly found in dry regions like Sudan, Deccan (India), or eastern Australia (Deckers et al., 2003), and (ii) Oxisols, which are found in humid Tropics, exhibit a large textural variability, and contain non-swelling clay (kaolinite) with much higher permeability than Vertisols (Spaargaren and Deckers, 1998). The Oxisols mapped as Clay in the Reynolds map and inducing a large negative evapotranspiration bias call for a better

representation of the Clay texture, with a soil texture map that distinguishes the two types of clays with different hydrologic behaviours. In contrast, neglecting Vertisols leads to overestimate evapotranspiration which is the case with the Zobler map in Deccan and Sudan (Fig. 8b,d), so the corresponding biases switch sign from negative to positive in Deccan, and become more positive in Sudan. These problems come from the simplification of the Zobler map in the ORCHIDEE model, which converts the original "very fine" soils to Clay Loam (section 2.2). Vertisols are also overlooked in Australia by the simplified Zobler map and by the SP-MIP map (Fig. 1), but with unsignificant impact on evapotranspiration in this strongly water-stressed area (Fig. 6). Finally, in Central America, the SP-MIP soil map shows a much higher presence of Clay compared to the Zobler and Reynolds soil maps. It should be underlined that the original 1km SoilGrids from which the SP-MIP map was derived does not show this dominance of Clay in this area, and we think that this feature is an error in the SP-MIP map. This over-representation of Clay turned the evapotranspiration bias from null/positive (with the Reynolds and Zobler maps) to negative.

### 3.5. Sensitivity of the simulated water budget to global soil texture maps at different scales

At the global scale like at the point-scale, the three complex soil texture maps result in very similar terrestrial water budgets (Fig. 10). Whichever the hydrologic variable, the global mean differences induced by these three maps (EXP2, EXP3 and EXP4) are smaller than the ones induced by different meteorological forcing (EXP1 vs EXP2), which are comparable to the uncertainty range between several observation-based estimates of the terrestrial water budget (Section 2.4). Compared to these estimates, it is also worth noting that ORCHIDEE simulates fairly well the mean partition between evapotranspiration and total runoff with any of the complex texture maps.

In contrast, the use of spatially uniform soil texture maps (EXP6 to EXP9) induces major differences in surface runoff, drainage and soil moisture. The strong decrease of soil moisture from EXP4 to EXP5 is not only due to the PTF change between these simulation, but more importantly to the relaxation of the decrease of $K_s$ with depth, which leads to larger $K_s$ at the bottom of the soil column, favouring drainage, thus reducing soil moisture. The global water budgets resulting from the uniform maps are in agreement with the response of the model to soil texture (section 3.2). In particular, the uniform clay map (EXP9) induces high soil moisture and surface runoff, and low drainage, compared to the other uniform maps, while the uniform coarse map (Loamy Sand in EXP8, but Sand would give similar results based on Fig. 3) shows the opposite behaviour. Eventually, using a uniform coarse or fine texture (EXP8 or EXP9) brings the simulated global mean evapotranspiration and runoff considerably out of the observed range, contrarily to the uniform medium texture maps (EXP6, EXP7). Overall, these uniform experiments tell us the maximum range of change we can expect from any kind of soil texture map change. For instance, the largest difference in mean global scale evapotranspiration (between the uniform clay and silt experiments owing to the non-monotonic response underlined in Figs. 3 and 4) is 0.1 mm/d, *i.e.* 8% of the global mean evapotranspiration using the complex soil texture maps and the same climate forcing.

To analyse the scale-related impact of soil texture maps on simulated fluxes, we upscaled the map of annual mean evapotranspiration difference (EXP2-EXP4) to coarser resolutions, from 1° to the global scale, by averaging the values of the

difference (Fig. 11). The resulting probability density functions are shown in Figure 12a, and Figure 12b-e shows how some metrics characterizing these distributions evolve with the averaging scale. The first noticeable impact of upscaling to coarser resolutions is the decrease of extreme evapotranspiration differences (Fig. 12b,d), leading to a less scattered distribution, also confirmed by the decreasing standard deviation (Fig. 12c). This figure shows that evapotranspiration difference follows a symmetrical distribution for the coarsest resolutions (above 5°), and starts showing a dissymmetric distribution below 5°, with a prevalence of negative values. This can also be seen in Figure 12d where the median of the evapotranspiration difference is all the more negative as the resolution gets finer. Thus, the strong impact of the soil texture map change that can be found locally (section 3.4) is mitigated at larger scales, and particularly at the global scale at which the terrestrial water budget shows a very weak sensitivity to the soil texture maps, even if it is statistically significant (Figs. 10 & 11).

## 4. Discussion and conclusions

Using the ORCHIDEE LSM and different soil texture maps, we found that the model shows a realistic sensitivity of surface runoff, drainage and soil moisture to soil texture compared to experimental and field studies (Rawls et al., 1993; Osman, 2013). These sensitivities lead to higher simulated evapotranspiration and lower total runoff for medium textures, which are discernable against other sources of variability when sorting the twelve USDA texture classes based on their median diameter. Apart in some areas which exhibit important differences in evapotranspiration, often attributed to the Clay texture class, the three complex soil texture maps tested here lead to similar water budgets at all scales, and the large uncertainties in observation-based products and climate forcing datasets make it impossible to conclude which map gives the best simulation.

These numerical results are specific to the ORCHIDEE model and the selected maps, but this model and these maps are representative examples of most state-of-the-art LSM applications (Vereecken et al., 2019), and comparable results were obtained with another LSM and other maps (De Lannoy et al., 2014). Besides, preliminary analyses of the LSM simulations conducted for the SP-MIP project (Gudmundsson & Cuntz, 2017) seem to confirm that varying soil parameters (resulting from different soil texture maps and different PTFs) have a small impact on long-term mean simulated evapotranspiration at the global scale, compared to other relevant uncertainties, including inter-model differences.

As mentioned in Introduction, much stronger responses to soil properties have been reported from bucket-type LSMs. It must be underlined, however, that these papers considered much larger changes of soil properties, which reduces in bucket-type models to available water holding capacity (AWC), combining information on porosity, soil depth, and the difference between field capacity and wilting point. As an example, the main changes discussed in Stamm et al. (1994), Ducharne & Laval (2000), de Rosnay & Polcher (1998), and Milly & Dunne (1994), correspond respectively to AWC changes of +75%, +110%, +200%, and +1400%, while the AWC changes when switching among the three soil texture maps used in the present paper does not exceed 5% (Table 2).

The weak sensitivity of the model to the three complex soil maps but in very specific areas is probably largely explained by their spatial similarity, which can be primarily attributed to their shared dependence on the FAO/UNESCO Soil Map, although weaker in SoilGrids thus in the SP-MIP map. Another reason is the coarse spatial resolution at which soil texture is used in ORCHIDEE and most LSMs, since selecting the dominant soil texture in every grid cell (here with 0.5° side, *ca*. 50 km) statistically enhances medium textures. As the latter lead to higher evapotranspiration and smaller total runoff than

more extreme textures (with larger percent of sand or clay particles), an important consequence, from a water budget point of view, is that dominant soil textures should favor excessive evapotranspiration and insufficient total runoff.

      Many alternative parameter upscaling methods were proposed to better preserve high resolution soil information, often based on averaging operators (usefully optimized to match coarse-scale observed streamflow in Samaniego et al., 2010), while Montzka et al. (2017) deduce upscaled parameters from theoretically upscaled hydraulic conductivity and diffusivity

curves. More invasive approaches would consist in describing the effects of high resolution soil information directly in the model equations, as frequently done for the effect of $K_s$ on infiltration owing to tractable statistical distributions (Vereecken et al., 2019). We lack similar developments for the full range of simulated water fluxes, apart from the partitioning of each grid cell into three soil columns with different soil textures, tested by de Rosnay et al. (2002) in ORCHIDEE but now abandoned.

      The soil texture maps themselves can also be questioned. When compared to the FAO soil order map (Fig. S4), the

SP-MIP map (following SoilGrids) tends to amplify the extent of sandy soils in Sahara and Saudi Arabia but ignores most sandy soils in Asia (e.g. Taklamakan desert). The largest evapotranspiration changes in our simulations were found in areas where the three soil texture maps disagree in their representation of clay soils, which calls for a better representation of this class in the soil texture maps. Of particular relevance is the distinction between Vertisols and Oxisols because of their very different hydrological properties. More generally, the use of simple PTFs based on soil texture classes only is increasingly

questioned. Firstly, they overlook the first-order influence of bulk density and soil structure, which require information on organic matter content (Smettem, 1987; Rahmati et al., 2018; Sun et al., 2018) and coarse fragments exceeding 2 mm, frequent in many soils (Brakensiek and Rawls, 1994; Valentin, 1994). Secondly, the simplifying assumption that soil texture is homogeneous vertically throughout the soil column should be revised. A particular attention should be paid on surface soil properties in areas prone to soil crusting (Valentin et al., 2008; Gal et al., 2017), which mainly include loamy soils (Rawls et

al., 1990) and also arid and semi-arid soils (Valentin and Bresson, 1992), producing high total runoff (Yair, 1990; Casenave and Valentin, 1992; Karambiri et al., 2003; Bouvier et al., 2018). Thus, using other sources of information than soil texture to derive the geographic distribution of soil properties may lead to clearer and broader improvements of the simulated water budget than the ones analysed here owing to mineral soil texture maps alone.

**Code availability**

The version of the ORCHIDEE model used for this study is based on tag 2.0, freely available from http://forge.ipsl.jussieu.fr/orchidee/browser/tags/ORCHIDEE_2_0/ORCHIDEE/

Small modifications were coded to read new maps of soil texture or soil parameters, and the corresponding code can be obtained upon request to first author.

**Data availability**

The GLEAM dataset used in this study can be freely accessed from www.GLEAM.eu. Primary data used in the analysis and other supplementary information that may be useful in reproducing the author's work can be obtained by contacting the corresponding author.

**Author contribution**

ST, AD and CV designed the research. ST performed the simulations, analysed the data and prepared a draft of the manuscript. All authors contributed to interpreting results, discussing findings and improving the manuscript.

**Competing interests**

The authors declare that they have no conflict of interest.

**Acknowledgments**

The ORCHIDEE simulations were performed using the IDRIS computational facilities (Institut du Développement et des Ressources en Informatique Scientifique, CNRS, France). Some of them were designed by Lukas Gudmundsson and Matthias Cuntz for the SP-MIP project.

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

     **Tables and Figures**

**Table 1. Summary of the experiments used in this study. Texture distribution displays the percentage of each soil texture in the used soil map.\* Indicates the experiments used in the SP-MIP. See Figure 1 for color codes.**

| Experiment | Soil map | Climate Forcing | PTF | Text.Distrib |
|---|---|---|---|---|
| EXP1 | Reynolds | CRU-NCEP | Carsel & Parrish (1988) | |
| EXP2 | Reynolds | GSWP3 | Carsel & Parrish (1988) | |
| EXP3* | Zobler | GSWP3 | Carsel & Parrish (1988) | |
| EXP4* | SP-MIP | GSWP3 | Carsel & Parrish (1988) | |
| EXP5* | SP-MIP | GSWP3 | Schaap et al. (2001) | |
| EXP6* | Loam | GSWP3 | Schaap et al. (2001) | |
| EXP7* | Silt | GSWP3 | Schaap et al. (2001) | |
| EXP8* | Loamy Sand | GSWP3 | Schaap et al. (2001) | |
| EXP9* | Clay | GSWP3 | Schaap et al. (2001) | |

     **Table 2. Percent sand, silt and clay contents of the geometric centroids of the 12 USDA soil texture classes. *dm*: the computed median diameter.**

| Texture class | Label | % Clay | % Silt | % Sand | $dm$ (μm) |
|---|---|---|---|---|---|
| Clay | C | 62.9 | 17.5 | 19.5 | 1.6 |
| Silty Clay | SiC | 46.7 | 46.7 | 6.7 | 5.4 |
| Silty Clay Loam | SiCL | 33.8 | 56.3 | 10.0 | 15.9 |
| Clay Loam | CL | 33.8 | 33.8 | 32.5 | 25.1 |
| Silt | Si | 5.3 | 87.3 | 7.3 | 26.6 |
| Silt Loam | SiL | 13.4 | 65.2 | 21.4 | 29.0 |
| Loam | L | 18.7 | 40.2 | 41.0 | 39.3 |
| Sandy Clay | SaC | 41.7 | 6.7 | 51.7 | 112.9 |
| Sandy Clay Loam | SaCL | 27.1 | 12.9 | 59.9 | 373.3 |
| Sandy Loam | SaL | 10.4 | 25.1 | 64.6 | 490.7 |
| Loamy Sand | LSa | 5.8 | 12.5 | 81.7 | 806.1 |
| Sand | Sa | 3.3 | 5.0 | 91.7 | 936.4 |

**Table 3. Percent overlap between the three tested soil texture maps.**

| | SP-MIP | Reynolds | Zobler |
|---|---|---|---|
| **SP-MIP** | 100.0 | | |
| **Reynolds** | 41.2 | 100.0 | |
| **Zobler** | 46.0 | 52.0 | 100.0 |
| **Unif. Loam** | 48.5 | 43.9 | 64.3 |
| **Unif. Silt** | 3.3 | 0.0 | 0.0 |
| **Unif. Loamy Sand** | 2.1 | 6.0 | 0.0 |
| **Unif. Clay** | 2.7 | 5.8 | 0.0 |

**Table 4. Statistical descriptors of the soil parameter maps corresponding to the three complex soil texture maps (excluding Antarctica Greenland): mean and standard deviation (SD) of each parameter map; and correlation coefficients between the three pair of maps.**

|  |  | SP-MIP | Reynolds | Zobler |
|---|---|---|---|---|
| **log(*dm*)** | Mean (log μm) | 4.48 | 4.23 | 4.25 |
|  | SD (log μm) | 1.51 | 1.65 | 1.15 |
|  | Cor. SP-MIP | 1.00 | 0.38 | 0.35 |
|  | Cor. Reynolds | 0.38 | 1.00 | 0.57 |
| **K$_s$** | Mean (mm/d) | 740 | 643 | 428 |
|  | SD (mm/d) | 1539 | 1261 | 376 |
|  | Cor. SP-MIP | 1.00 | 0.38 | 0.36 |
|  | Cor. Reynolds | 0.38 | 1.00 | 0.57 |
| **Saturated water content** | Mean (m$^3$/m$^3$) | 0.414 | 0.416 | 0.422 |
|  | SD (m$^3$/m$^3$) | 0.017 | 0.018 | 0.010 |
|  | Cor. SP-MIP | 1.00 | 0.40 | 0.22 |
|  | Cor. Reynolds | 0.40 | 1.0 | 0.35 |
| **Field capacity** | Mean (m$^3$/m$^3$) | 0.177 | 0.182 | 0.170 |
|  | SD (m$^3$/m$^3$) | 0.064 | 0.069 | 0.046 |
|  | Cor. SP-MIP | 1.00 | 0.41 | 0.36 |
|  | Cor. Reynolds | 0.41 | 1.00 | 0.58 |
| **Wilting point** | Mean (m$^3$/m$^3$) | 0.104 | 0.107 | 0.092 |
|  | SD (m$^3$/m$^3$) | 0.044 | 0.054 | 0.026 |
|  | Cor. SP-MIP | 1.00 | 0.42 | 0.36 |
|  | Cor. Reynolds | 0.42 | 1.00 | 0.58 |
| **AWC** | Mean (mm) | 146.7 | 150.2 | 156.5 |
|  | SD (mm) | 56.9 | 54.4 | 39.8 |
|  | Cor. SP-MIP | 1.00 | 0.34 | 0.31 |
|  | Cor. Reynolds | 0.34 | 1.00 | 0.42 |


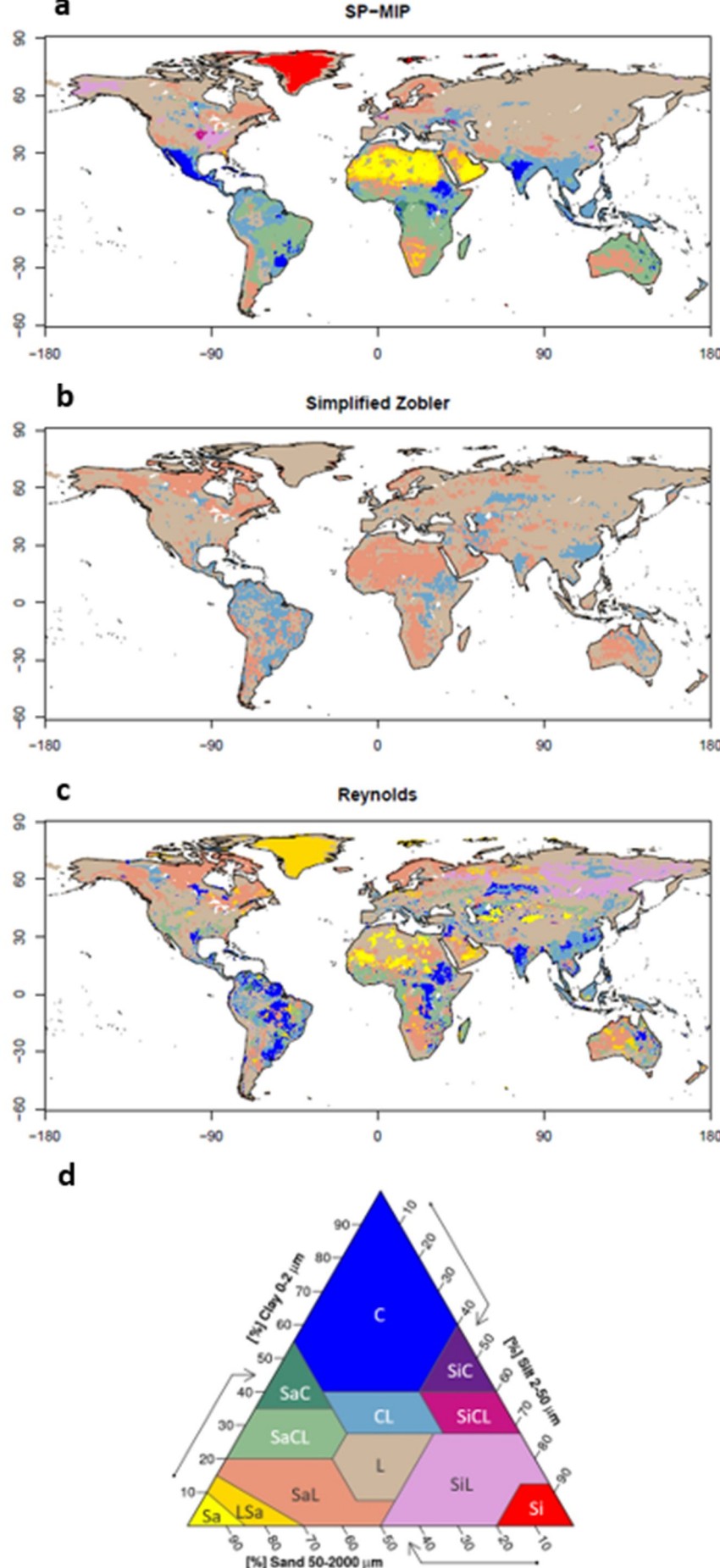

Figure 1. (a-c) Global maps of soil texture classes used in this study. (d) Soil texture triangle of the 12 textural classes as defined by the USDA. For texture labels see Table 2.

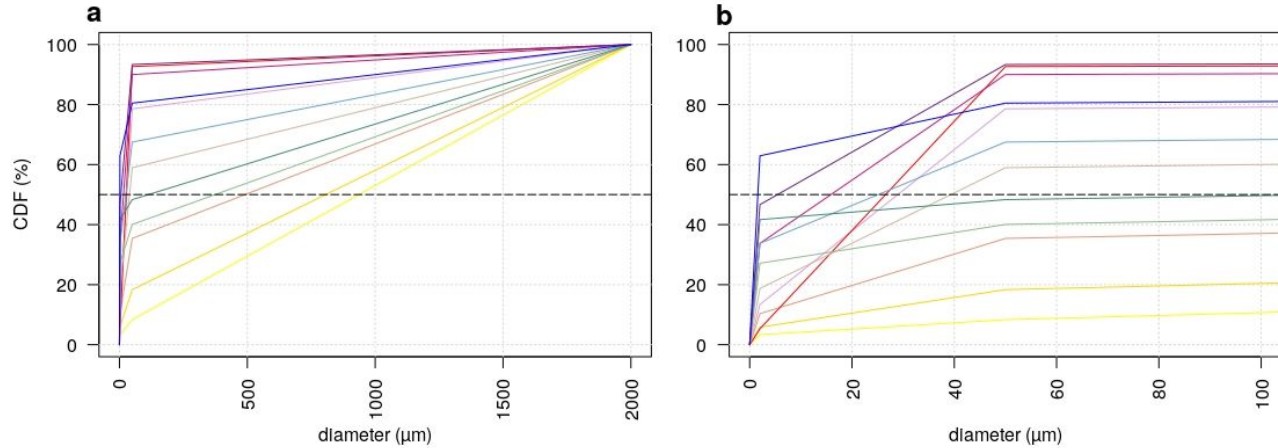

Figure 2. (a) Cumulative grain size distribution curves of the 12 USDA soil texture classes and (b) zoom over diameter interval [0,100 μm]. The dashed line defines the 50% cumulative value.

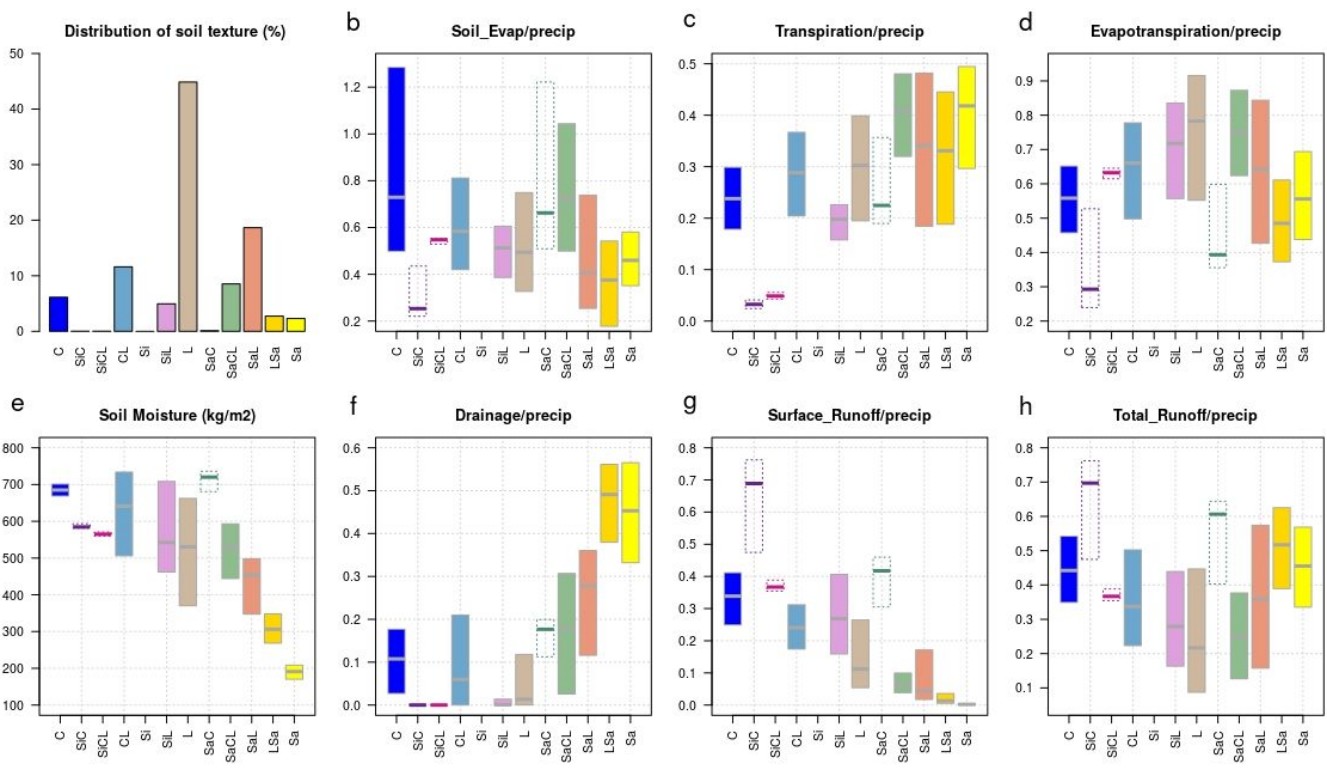

Figure 3. Variability of simulated variables of EXP2 over the land surface excluding Antarctica and Greenland, over the period 1980-2010, within each soil texture class. Soil texture classes are sorted from the finest to the coarsest based on dm (from left to right). See Figure 1 for color codes. Note that the Silt class is absent from Reynolds map. Dashed boxes correspond to texture classes covering less than 0.2% of the land area. Water fluxes are expressed as percent of mean precipitation. Soil moisture is averaged over areas with similar annual precipitation (between 1 and 2 mm/d), to remove impact of precipitation variation. Transpiration and soil evaporation fluxes are averaged over vegetated and bare soil fractions of the grid cells respectively.

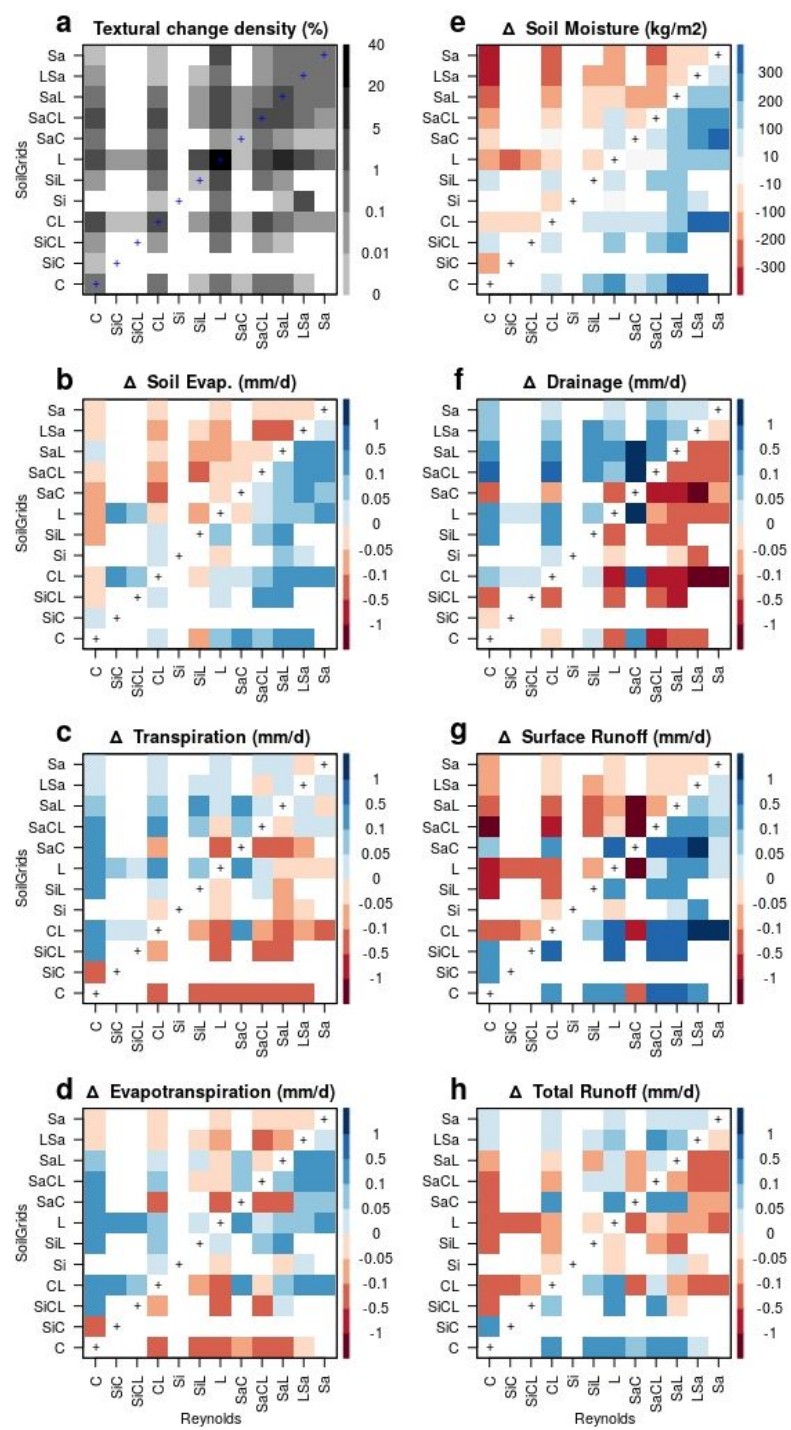

**Figure 4. Change in mean simulated variables over the globe land surface excluding Antarctica, averaged over the period 1980-2010, caused by changing the soil texture map from Reynolds to SP-MIP (EXP2 to EXP4). Soil texture classes are sorted from the finest (clay) to the coarsest (sand), in the x and y axis. The first plot illustrates the percentage of each textural change.**



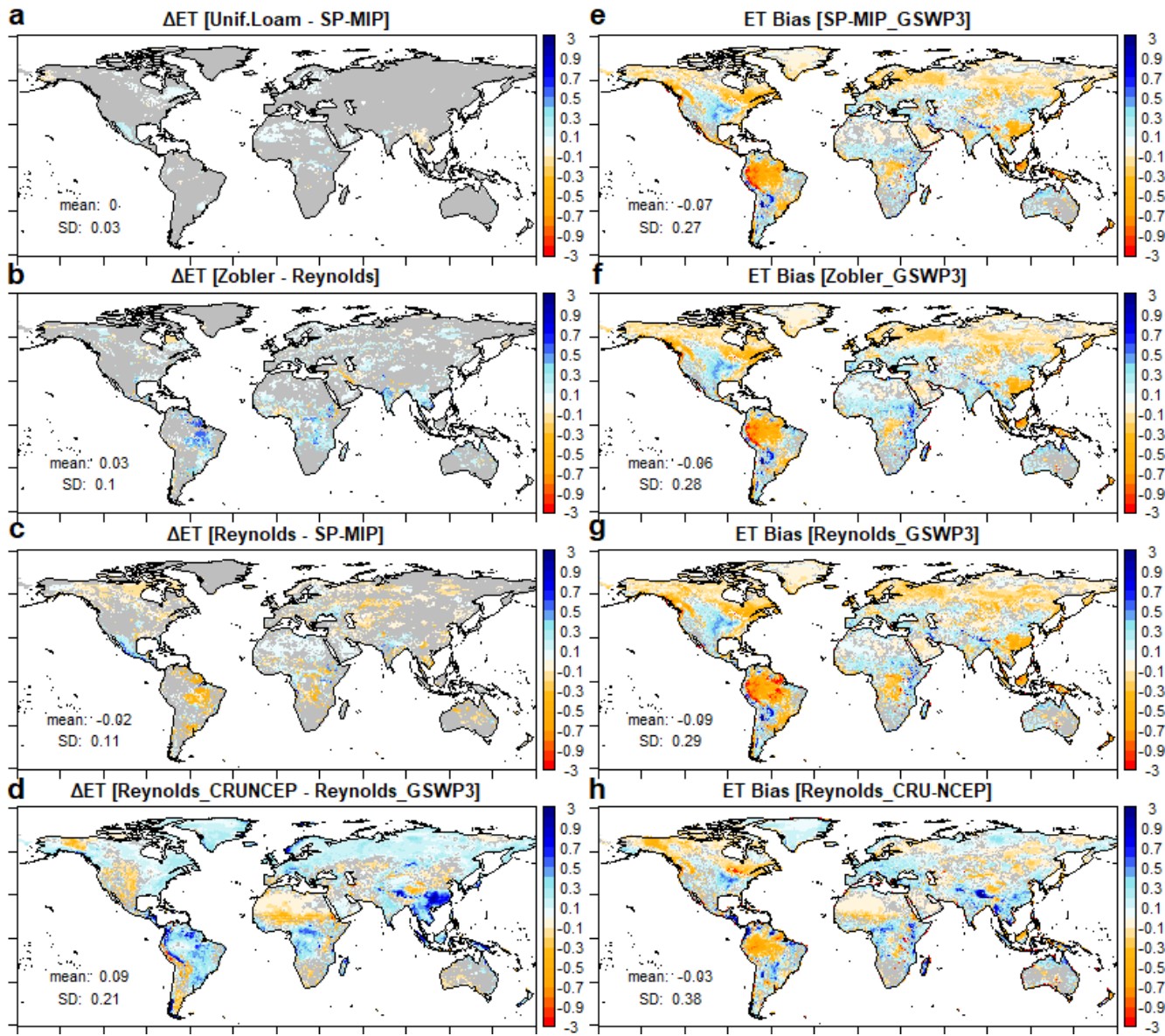

**Figure 5.** Spatial distribution of simulated annual mean evapotranspiration (averaged over 1980-2010): (left) differences between selected pairs of simulations (a: EXP6-EXP5, b: EXP3- EXP2, c: EXP2-EXP4, d: EXP1-EXP2); (right) biases with respect to GLEAM product (e: EXP4, f: EXP3, g: EXP2, h: EXP1). Grey color indicates that the difference is not statistically significant based on Student's t-test (with a p-value < 0.05). The printed means and standard deviation correspond to the full land area excluding Antarctica. Maps of GLEAM and simulated evapotranspiration of the 9 experiments are presented in Supplementary Figure S2.

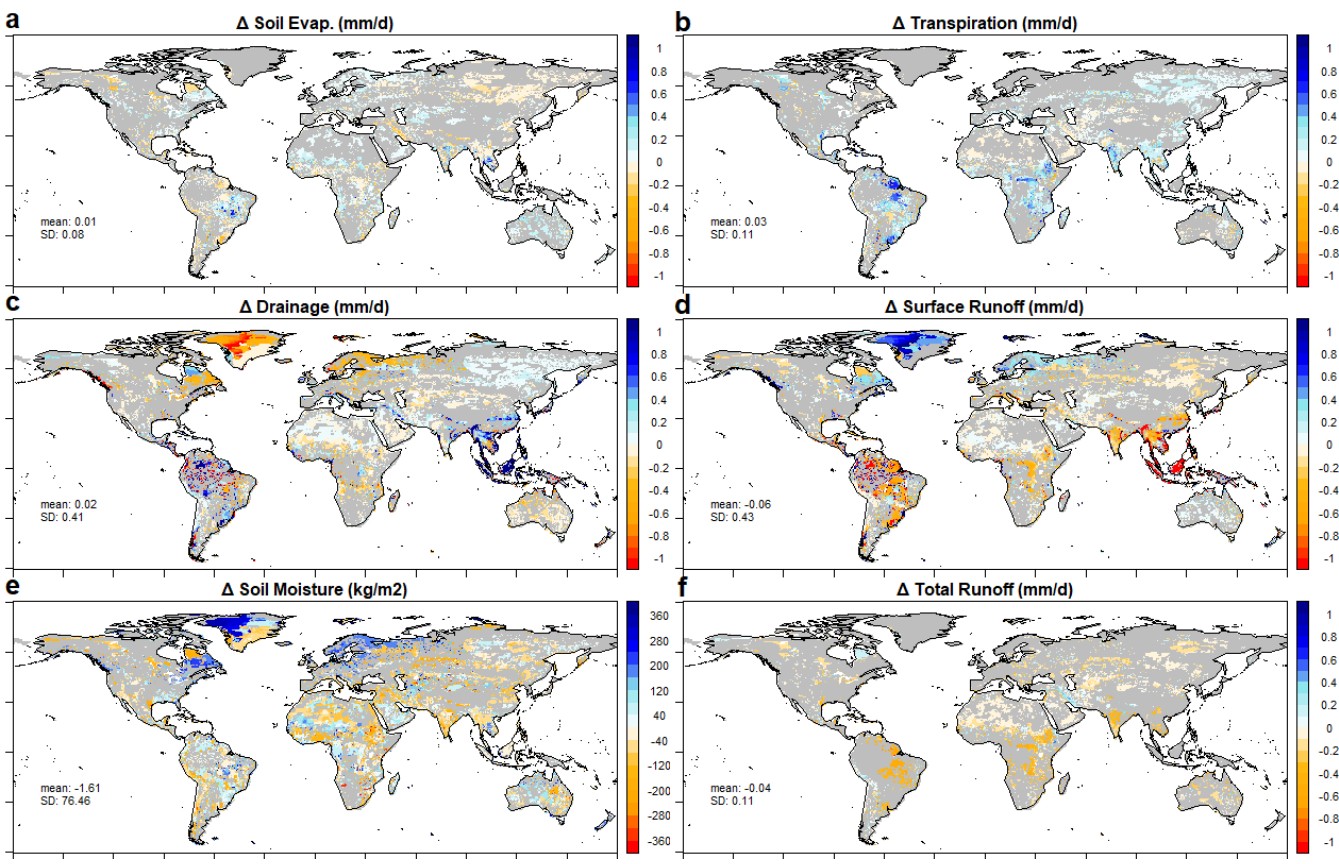


**Figure 6. Difference in simulated variables (averaged over the period 1980-2010) when Reynolds map is replaced by a Zobler map (EXP3 – EXP2). The corresponding difference for evapotranspiration is shown in Fig. 5b. Grey color indicates that the difference is not statistically significant based on Student's t-test (with a p-value < 0.05). Mean and standard deviation are averaged over the globe excluding Antarctica.**


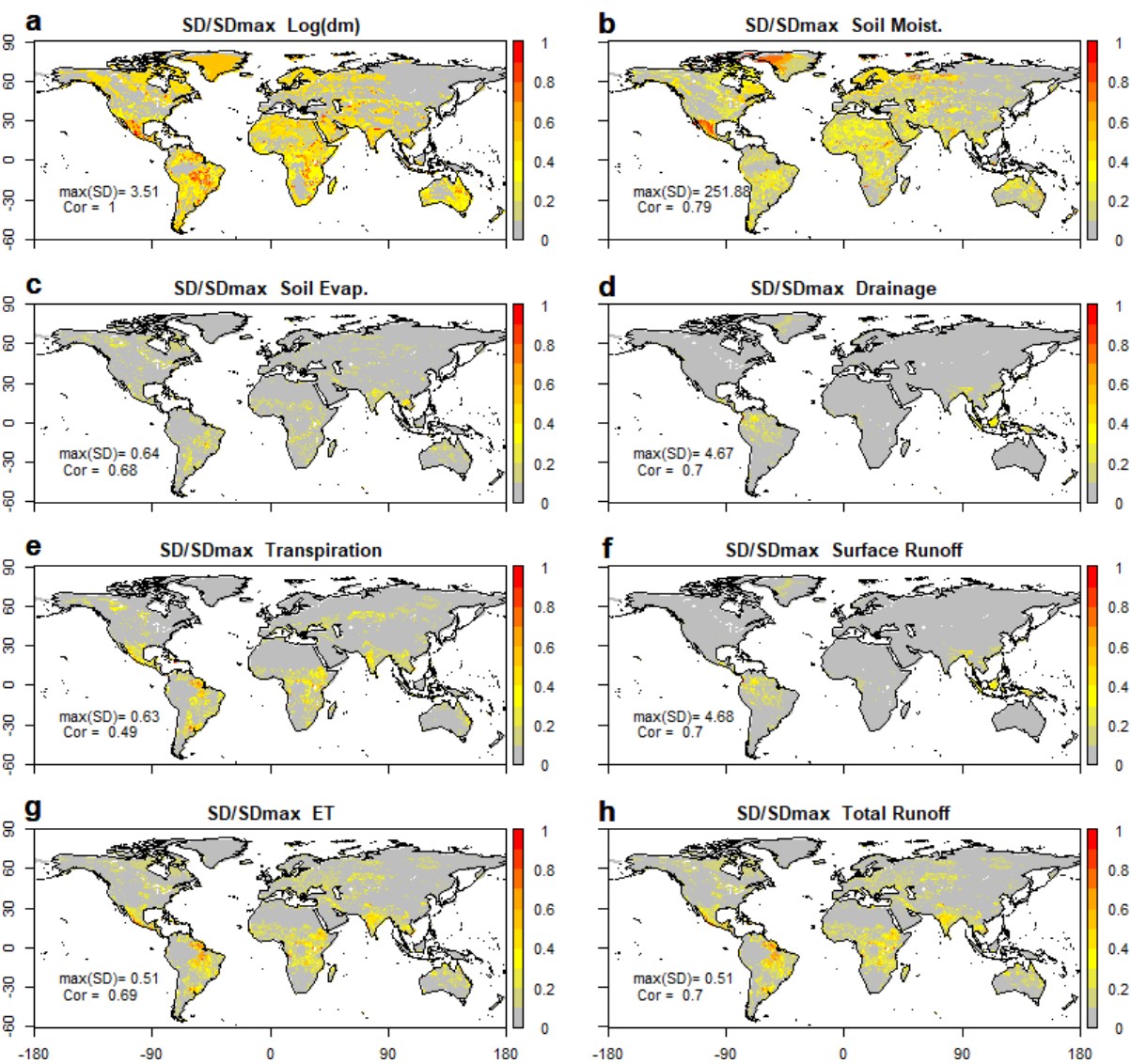

**Figure 7 Maps of the standard deviation (SD) of (a) the logarithm of median particle diameter (*dm*) given by the three complex soil texture maps (Reynolds, Zobler, SP-MIP), and (b-h) the mean annual simulated variables (in mm/d except for soil moisture in mm) using the three different maps. For easier comparison, each SD map is normalized by the maximum standard deviation of the map (maxSD), indicated in each map, with the spatial correlation coefficient (Cor) between the standard deviation of log(*dm*) and the standard deviation of each variable.**



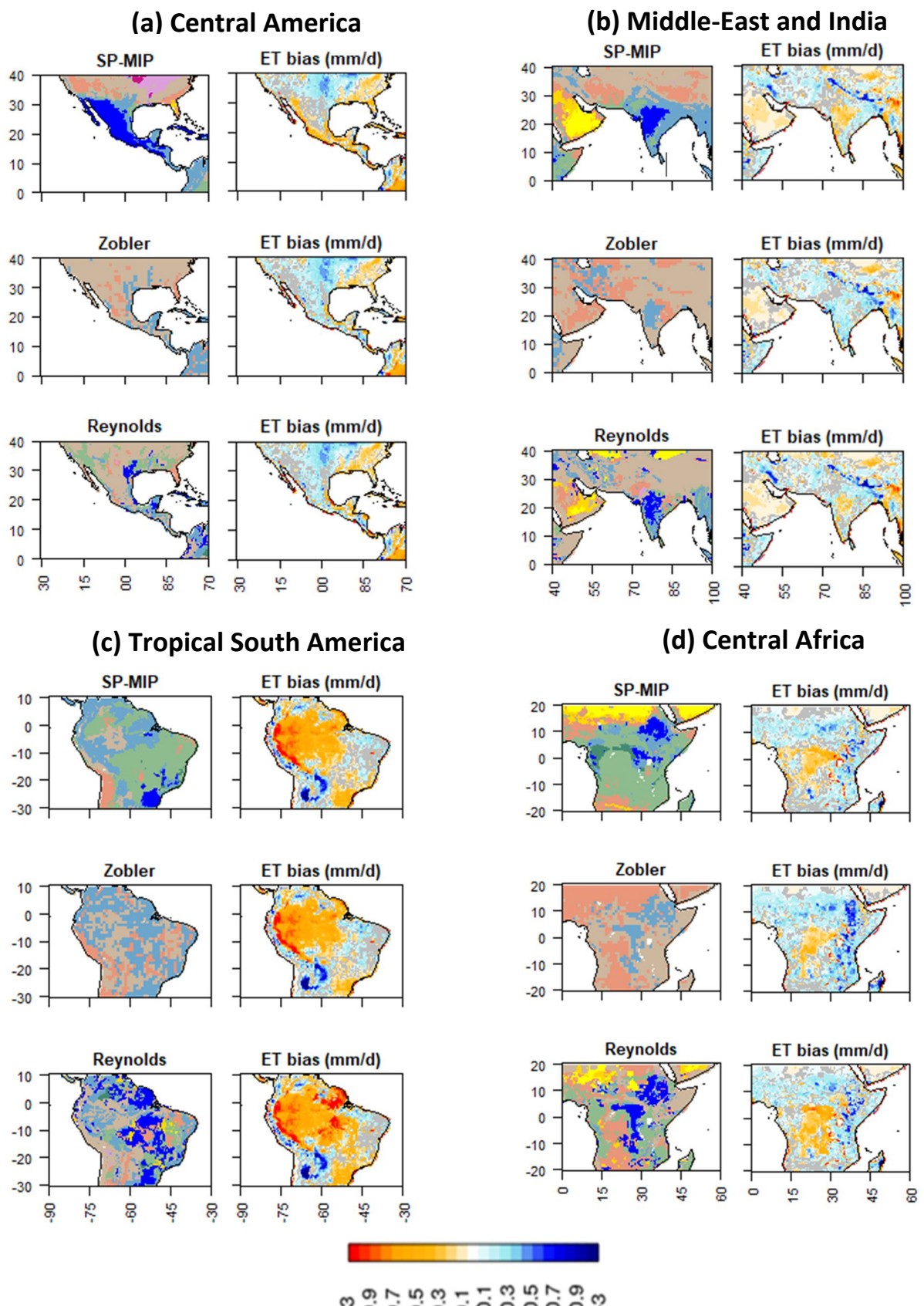

**Figure 8. Regional zooms on soil texture maps and the corresponding evapotranspiration bias maps (with respect to the GLEAM product) in four different areas. The colors scale on the right corresponds to the evapotranspiration bias maps, in which the grey color indicates that the bias is not statistically significant using Student's t-test with a p-value < 0.05. The colors of the soil texture maps are defined in Figure 1d.**

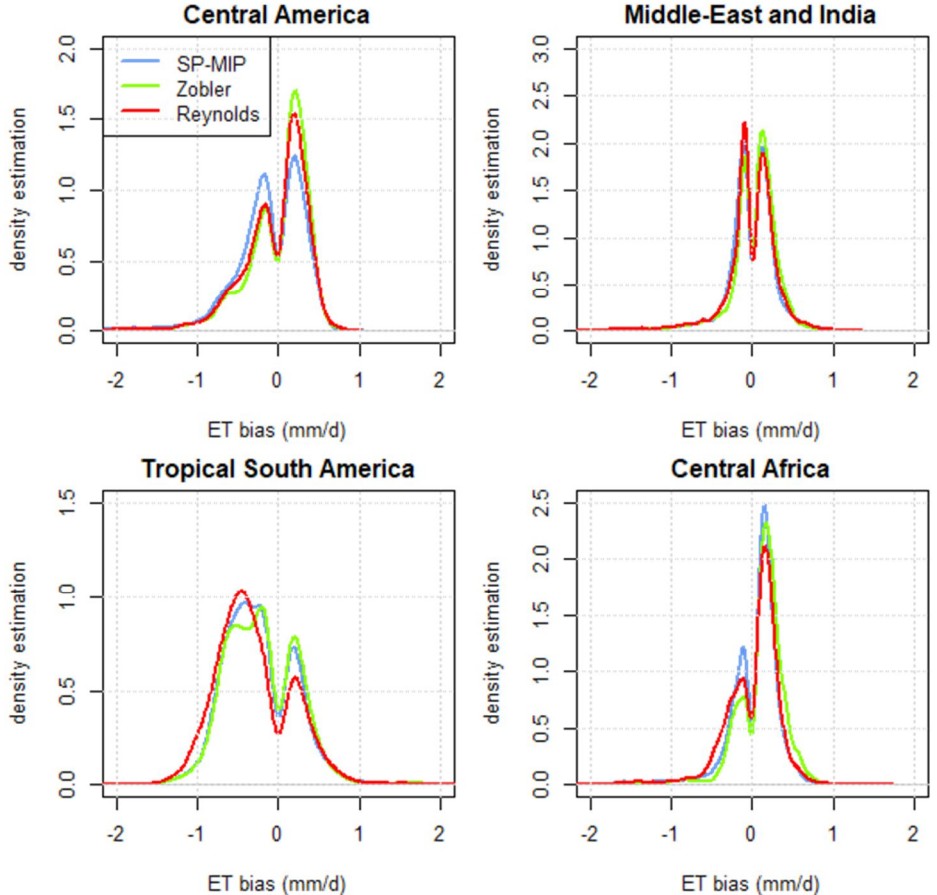

**Figure 9. Probability distribution of evapotranspiration bias in the 4 regions of Figure 8, for simulations EXP2, EXP3, EXP4 in red, green and blue respectively.**


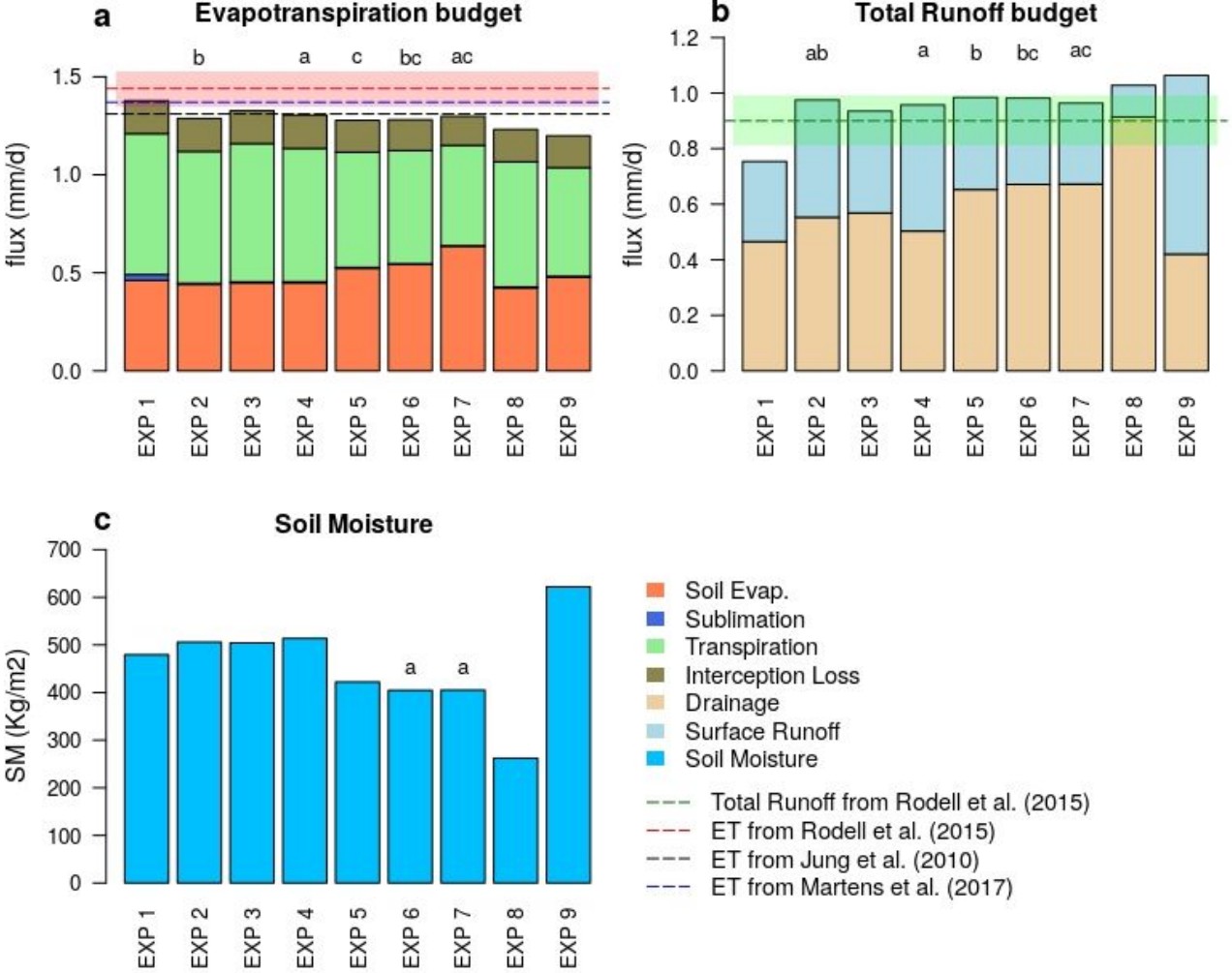

**Figure 10. Terrestrial water budget components for the nine simulations of Table 1, on average over 1980-2010 and over all land areas but Antarctica: (a) Evapotranspiration budget; (b) Total runoff budget; (c) Soil moisture. Letters above bars describe statistical significance: the mean difference between bars with the same letter is not statistically significant based on Student's t-test (with a p-value < 0.05). Red and green semi-transparent bands show the uncertainty range in the estimates of Rodell et al. (2015), for evapotranspiration and total runoff respectively. The estimated values of evapotranspiration and total runoff used for evaluation are described in section 2.4.**

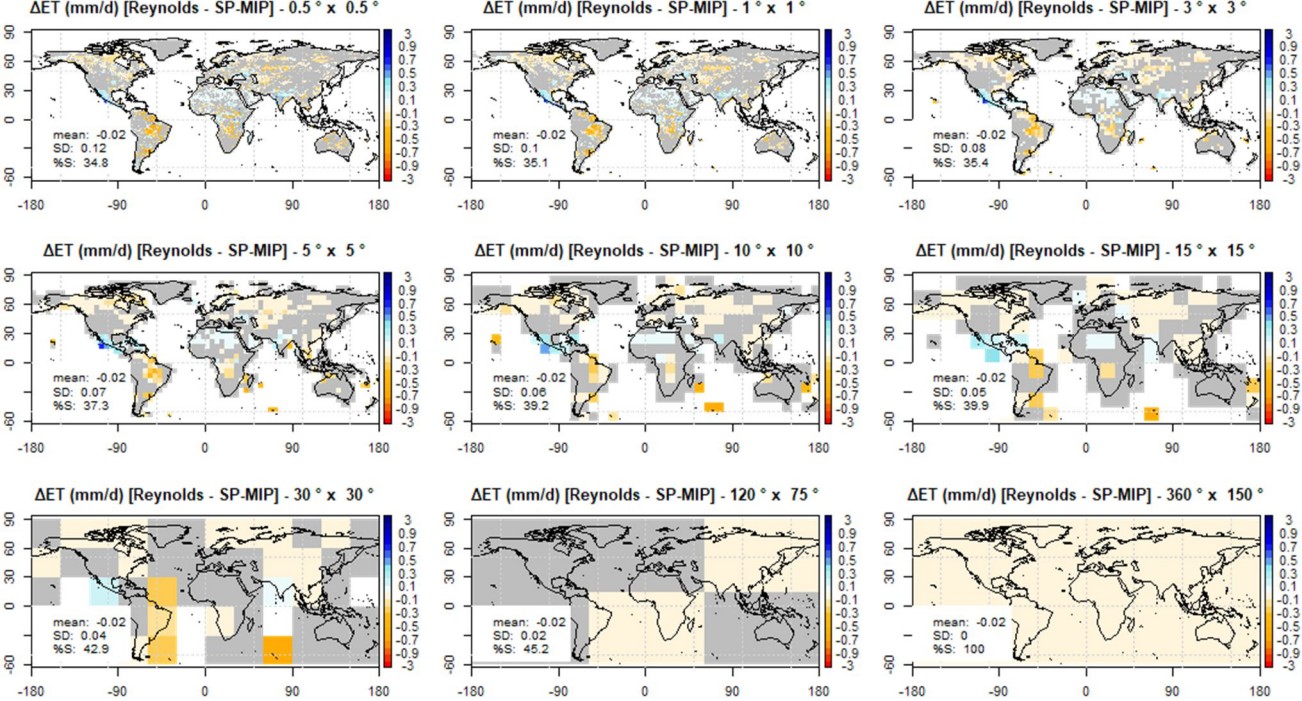

**Figure 11. Spatial distribution of simulated annual mean evapotranspiration: difference between EXP2 and EXP4 (Reynolds – SP-MIP), upscaled to different resolutions. Grey color indicates that the difference is not statistically significant at the tested resolution based on Student's t-test (with a p-value < 0.05). The printed means and standard deviations correspond to the full land area excluding Antarctica. %NS represents the percentage of land with non-significant differences.**


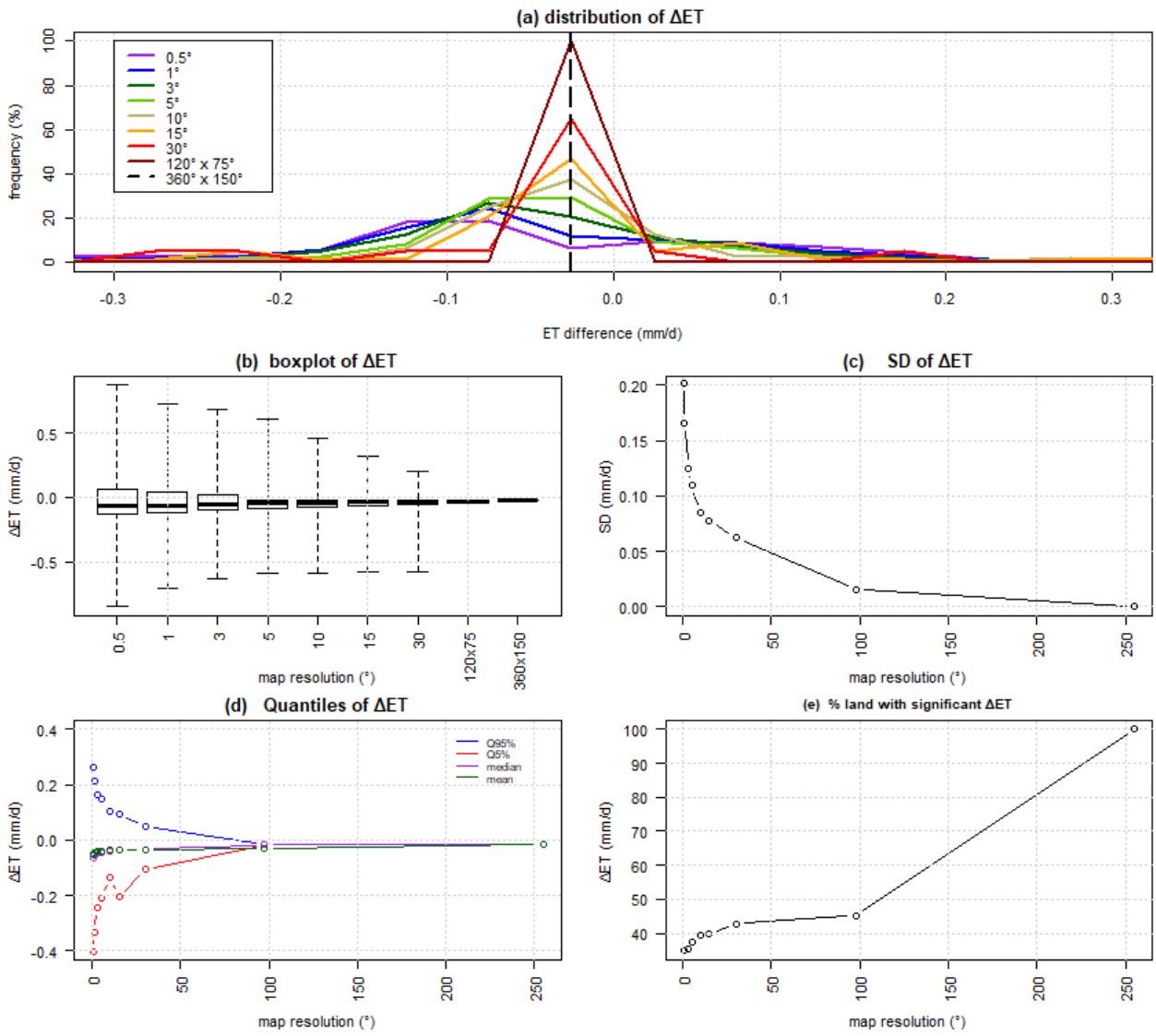

 **Figure 12. (a) Distribution of annual mean evapotranspiration difference (ΔET in mm/d) over land between EXP2 and EXP4, at different resolutions, (b-e) The corresponding statistical indicators (SD: standard deviation, and statistical significance assessed from a Student test with a p-value < 0.05).**