# Peer review of "Weak sensitivity of the terrestrial water budget to global soil texture maps in the ORCHIDEE land surface model"

_Hydrology and Earth System Sciences, 2019_

## Referee Comment (RC1) · Anonymous Referee #1 · 22 Sep 2019

Review of the manuscript "Weak sensitivity of the terrestrial water budget to global soil texture maps in the ORCHIDEE land surface model" by Tafasca et al. [hess-2019-305].

GENERAL COMMENTS

The paper by Tafasca et al. uses the ORCHIDEE land surface model to test the effect of using different soil texture maps on the water budget at the global scale and concludes that, given the similarities between the tested maps, the choice of input soil texture map is not crucial for large scale modeling (compared to the bias due to the choice of, for example, meteorological forcings). I think that the study of the impact of biases in the estimates in soil properties on water and energy budgets is of great importance for

assessing the accuracy of LSM simulations as well as to guide their parameterization. However, the manuscript by Tefasca and co-authors, in its current form, lacks a clear definition of its objectives and novelty. Most of the paper is devoted to showing the correct performance of a widely used model in modeling rainfall partitioning in different soil texture – this seems more a reality check for the model, than a novel analysis. The model is then used it to conclude that, given the similarity of the tested maps, at the scales under consideration the resulting bias in the hydrologic response is negligible (a result that could have been guessed without performing heavy numerical simulations). I believe the manuscript would better benefit from a more detailed (and quantitative) analysis of the relationship between the soil input bias and resulting hydrologic bias across scales, as detailed below.

MAJOR COMMENTS:

1. It is not clear what the novelty and the overall goal of the paper is. As it stands, it seems more of a modeling exercise using different soil maps, but without a clear scientific objective being proposed.

2. The authors use soil texture maps that are similar and conclude that they give similar results. If the soil maps are indeed not too different, how could the authors expect to observe any difference in the results (especially in terms of global fluxes where the main local differences are averaged out)? Along these lines, while the global/average water budget is similar, how different are the extremes (i.e., where the maps actually differ, what is the bias in the results)? In these terms, I think that a more detailed analysis of the biases induced in those areas where the maps differ would be more useful.

3. Lastly, at what scales do local differences in soil texture maps and the associated fluxes start to differ substantially? Can the authors define thresholds in these terms?

MINOR COMMENTS:

Abstract:

- Lines 10-11: "Here, we investigate the impact of soil texture on soil water fluxes and storage at global scale". What is the novelty here? The impact of having different soil texture (clay vs. sand) on infiltration/runoff partitioning is well known and a large scale application only seems a modeling exercise without added scientific value. I think the abstract (and paper) would benefit from the definition of a more precise research question/objective.

Introduction:

- Line 35: SoilGrids database is available at higher resolution too (250 m).

- In general, I think the Introduction lacks some clarity: It is not clear whether the focus here is on testing the LSM at the global scale, or on the effect of PTFs, or on the comparison of different soil texture maps. The paper would largely benefit from a more detailed introduction where the novelty and the goals of the paper are clearly defined in relation to state of the art knowledge on the subject.

Methods:

- Lines 66-67: at what depth are the soil texture maps? SoilGrids provides, for example, texture properties at different soil depths - why are the authors assuming an exponential decrease of Ks instead of evaluating it from textures at different depths?

- Lines 67-68: please provide a reference for both the exponential decrease with depth and the exponential distribution horizontally.

- Line 70: please provide references for the evapotranspiration model.

- Line 91: what is the error due to selecting only the dominant soil texture? Did the authors investigate the effect of upscaling by using some average (or weighted average) soil properties?

- Line 133: "network owing to machine learning" – please rephrase.

Results:

- Lines 133-135: the partitioning of rainfall in infiltration (soil moisture) and runoff differs among soil textures in a way that is well known and studied - I don't see the novelty here. Are the authors simply testing the model?

- Lines 153-154: "Switching. . .variables". If the maps are similar a priori, why would the authors expect any differences in the global water budget? It would probably be more useful, in my opinion, to focus on those areas where the maps are actually different and discuss the resulting biases in the hydrologic response in those areas.

- Lines 170-177: the results discussed here could have been expected without running massive simulations: the partitioning of rainfall in infiltration and runoff with different soil textures is well known. The exercise here seems more of a reality check for the model than some novel analysis.

- In general, I think the paper lacks a proper quantification of the differences between the soil texture maps and the related bias in simulated fluxes. If the authors could provide a clear quantitative link between the bias in soil maps and the resulting bias in hydrologic partitioning this would actually allow to extrapolate something from the analysis. As it stands, the analysis only seems a modeling exercise without any useful application. I believe it would be more impactful if the paper could provide answers to questions like: how much does the hydrologic response (e.g., runoff, infiltration, etc) change if the soil texture differs by a certain percentage? How do the probability distributions of the water budget components vary with the distributions of soil texture?

- The authors showed that hydrologic fluxes are more sensitive to changes in climatic forcings rather than soil texture maps. But how different are the climatic forcings used compared to soil texture maps? If the bias in the climatic forcing is, a priori, much higher, it is likely that the resulting hydrologic behavior will differ more.

Conclusions:

[Figure]

- Lines 197-198: the fact that the model has a realistic behavior should not be a main result. The orchidee model has been widely tested, and its ability to reproduce hydrologic fluxes properly in relation to different soil textures is not a novel result.

- Lines 210-212: What is the point of using spatially similar maps to see if they have any discernible effects on the hydrologic fluxes? If, a priori, the maps are similar, what is the point of the entire exercise?

- Line 214: Did the authors try to test some weighted average SHPs thus accounting for spatial variability instead of using the dominant soil texture in each cell?

- Line 225: A detailed analysis of the difference between the various maps should be given upfront. This only appears with Fig. 8 but it would be beneficial to have an in depth analysis of key differences among these maps (as well as of differences resulting from adopting different strategies for upscaling the higher resolution maps) at the beginning of the manuscript.

- The clay bias that is only briefly discussed in lines 225-230 seems actually a quite interesting point. If the prevalence of loamy texture in the texture maps is – in part – an artifact due to upscaling procedures and averaging, what would the bias be in the hydrologic partitioning if the actual texture in some grid cell was not as loamy as assumed?

- Lines 239-240: Some products (e.g., SoilGrids) have vertically variable information on soil texture – why didn't the authors use this information to relax the hypothesis of vertically homogeneous texture?

- Lines 236 – 244: most of the paper focused on soil texture, while only two PTFs were tested. Why is the conclusive paragraph of the manuscript on PTFs and inclusion of additional factors in currently used PTFs, while the manuscript only slightly touched this point? Although this is an interesting topic, I wouldn't embark into a discussion on PTFs at this point of the manuscript (as the authors didn't actually do an in depth

analysis of the bias induced by different PTFs).

---

## Referee Comment (RC2) · Anonymous Referee #2 · 5 Nov 2019

Review for Hydrology and Earth System Sciences Manuscript #: HESS-2019-305 Weak sensitivity of the terrestrial water budget to global soil texture maps in the OR-CHIDEE land surface model Authors: S. Tafasca, A. Ducharne and C. Valentin

Overall Comment This study investigates how a land surface model behaves against different global sets of soil parameters in terms of the terrestrial water balance. The experiment configurations follow the protocol of an ongoing international project, Soil Parameter Model Intercomparison Project. It concludes that the choice of the soil texture map is not crucial for large-scale modeling. The manuscript is well-written in a concise form, and their findings are important to our community. I encourage the HESS

journal to host this study, but the current version of the manuscript would not be at level to be accepted because of some hasty explanations and not enough interpretation and discussion.

Specific Comments # 13 : "medium texture" is not a clear term here. # 16 : Please provide reason or speculation why it "is not crucial". If not, it could mislead readers to consider soil parameters are not important, which is not true. # 81 : Please add data citation for GSWP3-v1 H. Kim. (2017). Global Soil Wetness Project Phase 3 Atmospheric Boundary Conditions (Experiment 1) [Data set]. Data Integration and Analysis System (DIAS). https://doi.org/10.20783/DIAS.501 # 93 : Rather "coarse and fine" than "medium and extreme"? # 133 : Please add the definition of "soil-moisture" which is sampled from each soil texture class which has a similar range of precipitation. Also, specify the sampling depth; top-soil, root-zone, full-column or any specific depth? # 142 – 145 : Please provide additional information how the model treats the root uptake and root-zone soil-moisture. Also, speculations on the role of groundwater capillary action would be a very important aspect, too. # 149 : How does leaf area index affect soil evaporation; interception loss, radiative transfer in canopy? Citing previous research would be helpful to show soil evaporation "strongly depends on other factors". # 158 – 163 : Only a part of Figure 4 has been touched. I suggest the authors to add in-depth interpretation of this figure. For example, the change of evaporation could be compared with of soil-moisture – (transpiration + total runoff). It is not recommended, but to discard this paragraph and Fig. 4 would be another option. # 175 : "coarse or clay" would be "coarse or fine" or "loamy sand or clay". # 175 – 177 : To me, evapotranspiration of EXP6 and EXP7 also seem out of the observed range. # 182 – 183 : Please specify regions. # 185 – 186 : To me, it does not seem to have a larger variability to the other fluxes (e.g., total runoff), particularly. # 209 : Please add "at the global scale".

Please also note the supplement to this comment:
https://www.hydrol-earth-syst-sci-discuss.net/hess-2019-305/hess-2019-305-RC2-

supplement.pdf

---

## Referee Comment (RC3) · Anonymous Referee #3 · 11 Nov 2019

Review of the manuscript hess-2019-305: "Weak sensitivity of the terrestrial water budget to global soil texture maps in the ORCHIDEE land surface model" by Tafasca et al.

General

This paper explores the impact of soil texture on the simulated water budget by the ORCHIDEE LSM at the global scale at 0.5 degree resolution. The authors conclude that the use of three different soil texture maps result in very similar terrestrial water budgets, and that the choice of the input soil texture map is not crucial for large scale modelling. While the study topic is very relevant and deserves publication, the manuscript needs to be revised. First, I think the authors should make in the Introduction their research question(s) clearer, in my opinion lines 52-53 are not sufficient, and it is also not quite clear why this research is different from earlier studies. The authors mention the use a physically based soil hydrological modelling component (including Richards equation) in these lines (line 53), but do not follow up in the Discussion and conclusions section. Furthermore, I believe the manuscript could benefit from a more detailed analysis and description on the differences between hydrological variables from different soil texture inputs (and PTFs), also focused on a regional/local scale. Finally, I am wondering why the authors did choose to scale up the high resolution soil texture dataset to the model resolution as a function of the dominant USDA soil texture class. Why not, when applicable, calculate the soil hydraulic parameters at high resolution, and then scale up (with appropriate scaling operators (for example in the line with Samaniego et al., 2010))?

Specific comments

A detailed description of the ORCHIDEE LSM would be helpful.

Line 13: explain "medium texture", I think not every reader knows what medium means

Line 13-14: "The three tested complex soil texture maps being rather similar by construction…". Do the authors mean that the soil texture maps are similar because of the way how they were constructed (taking the dominant USDA soil texture class)? As mentioned in the General comments, why not calculate soil hydraulic parameters at the high resolution scale? Indeed the soil texture maps are quite similar. Why then not focus more on sensitivity of PTFs (now two are used in this study)?

Line 35: 1-km SoilGrids database. A 250 m version is also available. Were the different soil layers also included in the analysis? And if yes, how? Also for example to calculate the exponential decline of Ks?

Lines 144-145: "Rather surprisingly, we find here…", Could you explain this in more detail? If drainage and transpiration decrease you would expect higher soil moisture values, right? The transpiration decrease is perhaps controlled by dominant vegetation type?

Lines 148-149: Could you elaborate more (also include references)? These factors should also affect evapotranspiration…

Line 158: Please describe Figure 4 in more detail.

Figure 5: EXP5 seems to show a large difference in soil moisture with EXP4. In my opinion an interesting result, but not mentioned in the text and explained.

Line 187-188 and Figure 7: Ok, indeed transpiration and soil evaporation show weak sensitivity, but other variables like drainage, surface runoff and soil moisture show a stronger sensitivity. For example, when you focus on Scandinavia, drainage decreases, surface runoff increases, and soil moisture increases. I believe the manuscript should also focus on these variables, in specific regions. Why is transpiration not affected here by the soil texture maps, and the water balance components as drainage, surface runoff and soil moisture do change?

Lines 208-209: what about other variables (water balance components) than evapotranspiration?

Lines 214-215: why not calculate hydraulic parameters at high resolution to remove some of that bias?

Lines 218-219: yes, the authors could have used these upscaling method of Samaniego.

Lines 226-235: again the focus on evapotranspiration. What about other water balance components?

Line 236-244: To include and end with this paragraph the authors should focus more on PTFs (methods and results) and describe these in more detail (methods).

---

## Author Comment (AC1) · 14 Jan 2020

**Weak sensitivity of the terrestrial water budget to global soil texture maps in the ORCHIDEE land surface model**

Salma TAFASCA[1] , Agnès DUCHARNE[1] , Christian VALENTIN[2]

**Reply to anonymous Referee #2**

**Overall Comment**

This study investigates how a land surface model behaves against different global sets of soil parameters in terms of the terrestrial water balance. The experiment configurations follow the protocol of an ongoing international project, Soil Parameter Model Intercomparison Project. It concludes that the choice of the soil texture map is not crucial for large-scale modeling. The manuscript is well-written in a concise form, and their findings are important to our community. I encourage the HESS journal to host this study, but the current version of the manuscript would not be at level to be accepted because of some hasty explanations and not enough interpretation and discussion.

We would like to thank the reviewer for taking the time to go through the paper, and for the relevant comments. According to all the three reviews that we received, we decided to make some substantial changes to the paper, in particular, the scientific question of the paper will be more clarified and new sub-sections will be added. A detailed presentation of the new structure of the paper is presented in the answer to Referee #1. In the following, we will provide a response to every point raised by Referee #2.

**Specific Comments**

**13 : "medium texture" is not a clear term here.**
Medium textures are the loamy textures, with medium dm (median diameter). To clarify this in the abstract, we will use the term loamy texture, which is clearer

**16 : Please provide reason or speculation why it "is not crucial". If not, it could mislead readers to consider soil parameters are not important, which is not true.**
The referred sentence relates to the soil texture maps and the not to the soil parameters. The specific reason is given by the previous sentence: "The three tested complex soil texture maps […] result in similar water budgets at all scales, compared to the uncertainties of observation-based products and meteorological forcing datasets". But this conclusion will be refined by underlining the areas where the choice of the soil texture map makes a significant difference, as detailed in a new subsection 3.4 "Regional zooms on greatly impacted areas", following the suggestion by Referee #1.

**81 : Please add data citation for GSWP3-v1 H. Kim. (2017). Global Soil Wetness Project Phase 3 Atmospheric Boundary Conditions (Experiment 1) [Data set]. Data Integration and Analysis System (DIAS). https://doi.org/10.20783/DIAS.501**
We will add this reference to the description of GSWP3-v1.

**93 : Rather "coarse and fine" than "medium and extreme"?**
Lines 92 and 93 will be changed to: "In addition, we tested four spatially uniform texture maps, corresponding to the Loam, Loamy Sand, Silt, and Clay texture classes (EXP6 to EXP9), to analyze the importance of spatial variability of soil texture on the global water budget."

**133 : Please add the definition of "soil-moisture" which is sampled from each soil texture class which has a similar range of precipitation. Also, specify the sampling depth; top-soil, rootzone, full-column or any specific depth?**
The simulated soil moisture corresponds here to the whole soil column (2m depth), as will be specified in the revised manuscript. As for the clustering by soil texture and normalization by mean precipitation, the latter is only used for the fluxes (see line 125), and not for soil moisture. We will add that this normalization is performed at point scale, using the pluri-annual mean of precipitation.

**142 – 145 : Please provide additional information how the model treats the root uptake and root-zone soil-moisture. Also, speculations on the role of groundwater capillary action would be a very important aspect, too.**
In ORCHIDEE, a root uptake function (describing the water extraction ability of roots) is used to calculate transpiration; it is a function of both the soil moisture profile and root density profile. The latter one follows an exponential decrease with depth at a rate depending on the plant functional type. We will add this in the model description, in section 2.1 of the new version of the paper.
  Regarding capillary rise, the standard version of ORCHIDEE used here considers free drainage at the soil bottom, which corresponds to the assumption of uniform soil moisture profile below the soil bottom, i.e. groundwater does not impact soil moisture through capillary rise. This will also be added in the model description section.

**149 : How does leaf area index affect soil evaporation; interception loss, radiative transfer in canopy? Citing previous research would be helpful to show soil evaporation "strongly depends on other factors".**
In ORCHIDEE, LAI has an important influence on the partition between soil evaporation and transpiration, via the fraction that is effectively covered by foliage, which increases exponentially with LAI with a coefficient of 0.5, also controlling light extinction through the canopy (Krinner et al. 2005). This fraction contributes to transpiration and interception loss, while the complementary fraction is assumed to be bare of vegetation, and only contributes to soil evaporation. This explanation will be added to the description of the ORCHIDEE LSM (section 2.1), thus complying with a request by Referee #3.
  To support the sentence of lines 148-149, we will also add the following references: Martens et al. (2017) and Wang et al (2018) regarding the anti-correlation between LAI and soil evaporation (further supported by the spatial correlation of -0.32 between these two variables in our simulation EXP2); the negative impact of vegetation on soil evaporation can also develop owing to the litter, which exerts a resistance to this flux (Ogée & Brunet, 2002; Sakaguchi & Zeng, 2009). However, the dependence of soil evaporation on climatic variables (temperature, potential ET) and soil moisture will not be expanded, as it is very well established.

**158 – 163 : Only a part of Figure 4 has been touched. I suggest the authors to add in-depth interpretation of this figure. For example, the change of evaporation could be compared with of soil-moisture – (transpiration + total runoff). It is not recommended, but to discard this paragraph and Fig. 4 would be another option.**

We agree with Referee #2 that Figure 4 was too briefly discussed. This figure is intended to show how the simulated variables change when only soil texture changes, to better analyze the model's response to the different soil textures. We propose to expand the last paragraph of section 3.1 addressing this Figure:

"By focusing this time on the point-scale changes induced by changing the soil texture map (from Reynolds to SoilGrids), Figure 4 highlights that the simulated soil evaporation decreases from fine to coarse textures, so that capillary retention, which is the main limiting factor to soil evaporation in ORCHIDEE, depends more strongly on soil moisture (higher for fine soils) than on intrinsic capillary forces (stronger for fine soils). We fail to see this behavior in Figure 3, which is likely due to the greater impact of diverse climatic conditions and vegetation associated with every soil texture. Figure 4 also confirms the results of Figure 3 for the other variables, including the decrease of soil moisture with coarser soils and the greater impact of soil texture on runoff variables (surface runoff and drainage). In particular, we find that replacing fine textures with coarse textures (above the first diagonal of the matrices) results in higher drainage (due to the higher permeability of coarse-textured soils) and lower surface runoff, with changes that can exceed 1mm/d in absolute value for some textural changes (all involving medium texture classes). As a result, less water is available in the soil, which leads to less soil evaporation, further leading to more transpiration (Fig. 4bc).

The convex behavior of total runoff with soil texture can also be seen in Figure 4h, which is antisymmetric along the two diagonals, thus defining four different kinds of total runoff change to soil texture change. This behavior results from the fact that total runoff sums up two variables of opposite response to soil texture change (surface runoff and drainage), the net response depending on the dominant component. Hence, changes to medium textures from either coarse or fine textures (left and right red triangles in Fig. 4h) lead to reduced total runoff, owing to reduced surface runoff in the first case, and reduced drainage in the second. In contrast, changes from medium texture to either coarse or fine textures lead to increased runoff (bottom and top blue triangles in Fig. 4h), owing to increased surface runoff or drainage, respectively. This pattern thus means that the medium textures correspond to the smallest total runoff. By means of long-term water conservation, the opposite patterns are found for total evapotranspiration changes (Figure 4d), because of the opposite responses of soil evaporation and transpiration to soil texture, and supporting the concave response of this flux to soil texture found in Figure 3."

**175 : "coarse or clay" would be "coarse or fine" or "loamy sand or clay".**

"coarse or clay" will be replaced by "coarse or fine".

**175 – 177 : To me, evapotranspiration of EXP6 and EXP7 also seem out of the observed range.**

It is true that land mean evapotranspiration of EXP6 is out of the observed range, but the one of EXP7 seems acceptable if we accept that the three estimates have an error margin, as shown for the estimates of Rodell et al (2015) for both total ET and total runoff, for which EXP6 and EXP7 fall within the confidence interval. Thus, when confronting the estimates of both mean total ET and total runoff over land, only EXP8 and EXP9 are clearly out of the

observed range. We will make this point clearer in the revised version of the paper.

**182 – 183 : Please specify regions.**
As explained in the response to Referee #1, the structure of the paper will be changed and a new sub-section dedicated to regional zooms on greatly impacted regions will be added, these regions include: Central America, Middle-East and India, Tropical South America and Central Africa. Lines 182 -183 will be changed accordingly.

**185 – 186 : To me, it does not seem to have a larger variability to the other fluxes (e.g., total runoff), particularly.**
We agree with the referee, but we are not comparing here the variability of evapotranspiration to the one of other variables. What we try to explain at lines 185-186 is the orange pocket in Fig 6a, with a decrease of ET when changing the soil texture to a uniform Loam, while we have previously shown that the medium textures correspond to the largest ET (cf. Figs 3 and 4, with the concave response of ET to soil texture). This region corresponds to a Clay Loam in SoilGrids (Fig. 1c), which is also found in many other regions (e.g. extensively in South America), without any significant change in ET when changing SoilGrids to the uniform Loam map. The underlying reason is thought to be the large variability of evapotranspiration within the Clay loam and Loam texture classes (Figure 3b), which makes it possible to have a local decrease of ET when changing soil texture from Clay loam to Loam despite the opposite relationship between the central values of these classes in Figure 3. The incriminated sentence will be rephrased based on the previous one, and moved to the new section of the revised version of the paper 3.4, as explained in answer to Referee #1.

**209 : Please add "at the global scale"**
It will be added in the new version of the paper.

**References:**

Krinner, G., Viovy, N., Noblet-Ducoudré, N. de, Ogée, J., Polcher, J., Friedlingstein, P., Ciais, P., Sitch, S. and Prentice, I. C.: A dynamic global vegetation model for studies of the coupled atmosphere-biosphere system, Glob. Biogeochem. Cycles, 19(1), doi:10.1029/2003GB002199, 2005.

Martens, B., Miralles, D. G., Lievens, H., Schalie, R. van der, Jeu, R. A. M. de, Fernández-Prieto, D., Beck, H. E., Dorigo, W. A. and Verhoest, N. E. C.: GLEAM v3: satellite-based land evaporation and root-zone soil moisture, Geosci. Model Dev., 10(5), 1903–1925, doi:https://doi.org/10.5194/gmd-10-1903-2017, 2017.

Ogée, J. and Brunet, Y.: A forest floor model for heat and moisture including a litter layer, J. Hydrol., 255(1), 212–233, doi:10.1016/S0022-1694(01)00515-7, 2002.

Rodell, M., Beaudoing, H. K., L'Ecuyer, T. S., Olson, W. S., Famiglietti, J. S., Houser, P. R., Adler, R., Bosilovich, M. G., Clayson, C. A., Chambers, D., Clark, E., Fetzer, E. J., Gao, X., Gu, G., Hilburn, K., Huffman, G. J., Lettenmaier, D. P., Liu, W. T., Robertson, F. R., Schlosser, C. A., Sheffield, J. and Wood, E. F.: The Observed State of the Water Cycle in the Early Twenty-First Century, J. Clim., 28(21), 8289–8318, doi:10.1175/JCLI-D-14-00555.1, 2015.

Sakaguchi, K. and Zeng, X.: Effects of soil wetness, plant litter, and under-canopy atmospheric stability on ground evaporation in the Community Land Model (CLM3.5), J. Geophys. Res. Atmospheres, 114(D1), doi:10.1029/2008JD010834, 2009.

Wang, F., Ducharne, A., Cheruy, F., Lo, M.-H. and Grandpeix, J.-Y.: Impact of a shallow groundwater table on the global water cycle in the IPSL land–atmosphere coupled model, Clim. Dyn., 50(9), 3505–3522, doi:10.1007/s00382-017-3820-9, 2018.

---

## Author Comment (AC2) · 20 Jan 2020

**Weak sensitivity of the terrestrial water budget to global soil texture maps in the ORCHIDEE land surface model**

Salma TAFASCA[1], Agnès DUCHARNE[1] , Christian VALENTIN[2]

**Reply to Anonymous Referee #1**

We sincerely thank the reviewer for taking the time to review our manuscript and for the very constructive comments he/she provided. We provide below a point-by-point response to these comments, numbered from C1 to C26 for convenience.

**C1:** The paper by Tafasca et al. uses the ORCHIDEE land surface model to test the effect of using different soil texture maps on the water budget at the global scale and concludes that, given the similarities between the tested maps, the choice of input soil texture map is not crucial for large scale modeling (compared to the bias due to the choice of, for example, meteorological forcings). I think that the study of the impact of biases in the estimates in soil properties on water and energy budgets is of great importance for assessing the accuracy of LSM simulations as well as to guide their parameterization.

However, the manuscript by Tafasca and co-authors, in its current form, lacks a clear definition of its objectives and novelty. Most of the paper is devoted to showing the correct performance of a widely used model in modeling rainfall partitioning in different soil texture – this seems more a reality check for the model, than a novel analysis. The model is then used it to conclude that, given the similarity of the tested maps, at the scales under consideration the resulting bias in the hydrologic response is negligible (a result that could have been guessed without performing heavy numerical simulations).

I believe the manuscript would better benefit from a more detailed (and quantitative) analysis of the relationship between the soil input bias and resulting hydrologic bias across scales, as detailed below.

We mostly agree with the analysis, and we propose to augment the paper with additional analyses, as suggested by the reviewer. The revised version of the paper will have the following structure, with proposed changes highlighted in italic:

1. Introduction
   *New paragraph at the end clarifying the scientific objectives of the paper and outlining the structure of the paper.*
2. Materials and methods
   2.1. Soil texture in the ORCHIDEE LSM
   2.2. Simulation protocol
   2.3. Calculation of median diameter *dm* for each of the 12 USDA soil texture classes
   2.4. Evaluation datasets
3. Results
   *3.1. Comparison of the tested soil texture maps*
      *This new subsection will detail the similarities and differences between the three tested soil maps. A detailed description of this section is found in C22.*

3.2. Point scale sensitivity to the 12 USDA texture classes
This section corresponds to the original section 3.1 of the submitted version of the paper, mostly unchanged.

3.3. Spatial patterns of simulated fluxes and ET bias
This subsection will analyze Figures 6 and 7 of the submitted paper, which correspond to paragraphs 2 and 3 of section 3.2 of the submitted version. *We will also add and discuss the new Figure A5, which is described in C17.*

3.4. *Regional zooms on greatly impacted areas*
*This subsection will highlight areas with important differences induced by the soil map changes (Figure A1). It will also include the discussion on clays found the closing section of the submitted manuscript*

3.5. *Sensitivity of the simulated water budget to global soil texture maps at different scales*
In addition to analyzing the global water budgets resulting from the different soil texture maps, *we propose to add an analysis of the impact of the upscaling resolution on the simulated water fluxes (see C5).*

4. Discussion and conclusions
Most of the discussion about clays will be removed (transferred to subsection 3.4), and *the main conclusions will be rephrased to fit the updated results. In particular, the weak sensitivity of the simulated water fluxes to the prescribed soil texture maps is mostly found at coarse scales (global water cycle), but the texture maps can have a large impact at small scales.*

The propositions detailed in the following would lead to add 5 new figures, and change one figure (Fig. 8) into 2 tables, thus leading to a total of 12 figures and 4 tables.

MAJOR COMMENTS:
**C2:** 1. It is not clear what the novelty and the overall goal of the paper is. As it stands, it seems more of a modeling exercise using different soil maps, but without a clear scientific objective being proposed.
The main objective of the study was and remains to examine the impact of various soil texture maps on the simulated hydrologic fluxes, from grid-point to continental scales. As discussed below, there is no consensus on what is the "best" soil texture map for global LSM applications, and the identification of the "best" soil map is thus an important scientific question for land surface modelers.
    Given our results and the bibliography, we can postulate at least two reasons for this lack of consensus on the "best" global soil texture map: (1) There is no paper trying to identify it; (2) The overall impact of changing the soil texture map on the simulated fluxes is quite small apart from specific areas. This weak sensitivity is probably a reason why there has been no publication on this topic until now, apart from De Lannoy et al., (2014), who document the improvement of one specific global soil texture map.
In this framework, the result we chose to focus on in the submitted manuscript was the weak sensitivity of the simulated water budget to the tested maps, because we felt it had useful practical consequences. Indeed, it means that the choice of the soil texture map, among the ones that are routinely available, is not a major issue for global scale modeling. The choice of the meteorological forcing dataset, for instance, has a much stronger impact. Yet, we perfectly understand the reviewer's point of view regarding the interest of identifying where the soil texture maps do matter, and we will add a specific sub-section of the Results (3.4) on this point.
As for the scientific goals of the paper, they will be clarified in an expanded paragraph at the end of the Introduction, given in response to comment C8.

**C3:** 2. The authors use soil texture maps that are similar and conclude that they give similar results. If the soil maps are indeed not too different, how could the authors expect to observe any difference in the results (especially in terms of global fluxes where the main local differences are averaged out)?

The reviewer is right, but the similarity of the soil texture maps is not *a priori* knowledge, as might be suggested by the abstract: it is actually an outcome of our study. At first sight, it is not straightforward that the three tested soil maps are similar (Fig. 1 of the paper), nurturing the question of the "best" soil texture map. It is after getting our results that we analyzed the similarities between the different soil maps to understand why the large-scale simulated fluxes are largely similar. Based on reviewer's suggestion C22, we propose to add a new sub-section in the Results (3.1) where we analyze the similarities and differences between the soil texture maps. We will also rephrase the abstract.

**C4:** Along these lines, while the global/average water budget is similar, how different are the extremes (i.e., where the maps actually differ, what is the bias in the results)? In these terms, I think that a more detailed analysis of the biases induced in those areas where the maps differ would be more useful.

We agree, and in the revised version, we will add a dedicated subsection (3.4). We propose to add Figures A1 and A2 below, where we zoomed on four 40°x60° areas where the ET bias is greatly impacted by soil texture map. Based on Figs. 3 and 4 of the submitted manuscript, the largest changes in ET and total runoff are expected where soil texture changes between medium and extreme (Clay or Sand) textures. Hence, the absence of Clay in the Zobler map results in important increase of ET bias (Fig. A1 and Fig. A2). In contrast, the Oxisols mapped as Clay in the Reynolds map correspond to a large negative ET bias (e.g. South America in Fig. A1 and Fig. A2). Another example is found in Central America, where the SoilGrids texture map provided by the SP-MIP team shows a large amount of Clay, which turns the ET bias from positive (with the Reynolds and Zobler maps) to negative. It must be underlined that the original 1km SoilGrids does not show this dominance of Clay in Mexico, and we think that this feature is an error of the SP-MIP map. Since we noticed some non-negligible differences between the original SoilGrids map and the one provided by SP-MIP which is used in this study, we decided to refer to the latter by the SP-MIP map rather than the SoilGrids map. This will be changed and clarified in the revised version of the paper.

The text describing Figures A1 and A2 in the new subsection will be based on the one already present in the Discussion of the submitted version, but without any supporting figure, which shows the importance of better describing the Clay texture, and calls for a soil texture map that distinguishes the two clay types which have different hydrologic behaviors: the Oxisols and the Vertisols. The other extreme soil texture (Sand) is mostly found in arid areas where water is a limiting factor, so the soil map change does not greatly impact the ET bias. It is the case in the Arabian Peninsula and the Sahara, where the sandy soils mapped in SoilGrids are absent in Zobler and only weakly present in Reynolds, but the ET bias hardly changes and remains negative.

[Figure]

*Figure A1 Local zooms on soil texture maps and the corresponding ET bias maps (with respect to the GLEAM product) in four different areas. The colors scale on right corresponds to the ET Bias maps, in which the grey color indicates that the bias is not statistically significant using Student's t-test with a p-value of 5%. The colors of the soil texture maps are defined in Figure 1d.*

[Figure]

*Figure A2 Probability distribution of ET bias in the 4 regions of Fig.A1, for simulations EXP2, EXP3, EXP4 in red, green and blue respectively.*

**C5:** 3. Lastly, at what scales do local differences in soil texture maps and the associated fluxes start to differ substantially? Can the authors define thresholds in these terms?

We agree that the impact of the scale of analysis on the simulated fluxes is an interesting point to look at. To this end, we decided to add a new sub-section 3.5 in the Results, called: "Sensitivity of the simulated water budget to global soil texture maps at different scales". To analyze the scale-related impact of soil texture maps on simulated fluxes, we reproduced Figure 6c of the submitted paper, but upscaled this map of annual mean ET difference (EXP2-EXP4) to coarser resolutions, from 1° to the global scale, by averaging the values of ET (Fig. A3). The resulting probability density functions (pdfs) are shown in Figure A4, and Figure A5 shows the evolution of some metrics characterizing these distributions with the averaging scale. The first noticeable impact of upscaling ET to coarser resolutions is the decrease of extreme ET differences (Fig. A5a,c), leading to a less scattered distribution, also confirmed by the decreasing standard deviation (Fig. A5 b).

These figures (which may be combined in one in the revised manuscript) show that ET follows a nearly normal distribution for the coarse resolutions (above 5°), and starts showing a dissymmetric distribution for the finest resolutions (below 5°), with a prevalence of negative values (Fig A4). This can also be seen in Figure A5c where the median value of the ET difference moves to more negative values as the resolution gets finer. As a consequence, if we wanted to define a threshold at which resolution starts to impact the distribution of annual mean ET, it would be the

5° resolution. We propose to include Figure A3, A4 and A5 as well as the aforementioned analysis in sub-section 3.5, in order to bridge the gap between the point-scale maps at which some strong impacts of the soil texture maps can be found regionally, and the global scale, at which the terrestrial water budget shows a very weak sensitivity to the soil texture maps, even if   they are statistically significant (Fig. 5).

Nonetheless, we would like to point out that this analysis is not exhaustive, as a thorough analysis of the impact of the soil texture map resolution on the simulated fluxes would require performing additional simulations with soil texture maps upscaled to different resolutions. This kind of analysis is out of the original scope of our paper, especially given the general trend in land surface modelling for always higher resolutions (Bierkens et al., 2015; Wood et al., 2011).

[Figure]

*Figure A3 Spatial distribution of simulated annual mean evapotranspiration: difference between EXP2 and EXP4 (Reynolds - SoilGrids), upscaled to different resolutions. Grey color indicates that the difference is not statistically significant at the tested resolution based on Student's t-test (with a p-value of 5%). The printed means and standard deviation correspond to the full land area excluding Antarctica. %NS represents the percentage of land with non-significant differences.*

[Figure]

*Figure A4 Distribution of annual mean ET difference between EXP2 and EXP4, at different resolutions. These distributions correspond to the maps of figure A3.*

[Figure]

*Figure A5 Statistical indicators of distribution of annual mean ET : difference between EXP2 and EXP4, at different resolutions*

MINOR COMMENTS:

Abstract:
**C6:** - Lines 10-11: "Here, we investigate the impact of soil texture on soil water fluxes and storage at global scale". What is the novelty here? The impact of having different soil texture (clay vs. sand) on infiltration/runoff partitioning is well known and a large scale application only seems a modeling exercise without added scientific value. I think the abstract (and paper) would benefit from the definition of a more precise research question/objective.

*We agree that our research questions were not well defined, as already discussed in C1. In the revised version of the paper, we will make a clearer description of the research questions in both the abstract and the introduction (cf. C8). The main conclusions will also be updated to match the revised manuscript.*

Introduction:
**C7:** - Line 35: SoilGrids database is available at higher resolution too (250 m).
*We will mention the availability of the 250m version of SoilGrids (Hengl et al., 2017) at line 35.*

**C8:** - In general, I think the Introduction lacks some clarity: It is not clear whether the focus here is on testing the LSM at the global scale, or on the effect of PTFs, or on the comparison of different soil texture maps. The paper would largely benefit from a more detailed introduction where the novelty and the goals of the paper are clearly defined in relation to state of the art knowledge on the subject.

*The main point of the paper is testing three different soil texture maps, broadly used by LSMs, and comparing their resulting hydrologic variables. In the revised version of our paper, this will be clarified by expanding the last line of the introduction to a more classical paragraph detailing the specific research question of the paper and the structure of the paper:*

*"Here, we aim at exploring more systematically the impact of soil texture on the water budget from point to global scale, using a state-of-the-art LSM with physically-based soil hydrology, and multiple input soil texture maps. After presenting the model and soil texture maps used in this work, the results are presented as follows. We first provide an analysis of the similarities and differences between the different soil maps, then, we evaluate the point-scale response of the model to different soil textures to make sure it displays a reliable behavior. This point-scale response is then analyzed from a geographic point of view, with a comparison to a distributed observation-based ET product, and a focus is made on areas with a large sensitivity to the soil texture maps. We finally explore how the magnitude and significance of the simulated ET changes with the scale of analysis up to the land scale, defining the terrestrial water budget. The closing section summarizes the main conclusions of the study, and discusses its limitations and perspectives. "*

Methods:
**C9:** - Lines 66-67: at what depth are the soil texture maps? SoilGrids provides, for example, texture properties at different soil depths - why are the authors assuming an exponential decrease of Ks instead of evaluating it from textures at different depths?

*We thank the reviewer for pointing out this non-stated information. SoilGrids is available at 7 different depths: 0cm, 5cm, 15cm, 30cm, 60cm, 100cm and 200cm. The SoilGrids map used in this study is the one at 0cm depth, as processed for the SP-MIP project. The Reynolds soil texture map*

is available at two different depths: 30cm and 100cm, and the first depth is used in this study. The Zobler map is available at one soil depth of 30cm. We will add this information in section 2.2 of the revised version of the paper.

Although some soil maps provide soil textures for different horizons, this information cannot be used in ORCHIDEE, as will be explained in the model description, in the revised version of the paper (cf. response to comment C6 Referee #3): *"Soil texture is assumed to be uniform over the soil column in ORCHIDEE, which does not permit to distinguish several soil horizons. However, $K_s$ decreases exponentially with depth, to account for the effects of soil compaction and bioturbation, as introduced by d'Orgeval et al. (2008) following Beven & Kirkby (1979)."* We also underline that the simplifying hypothesis of a uniform texture over the whole soil column is discussed in the concluding section of the submitted manuscript (lines 239-240).

**C10:** - Lines 67-68: please provide a reference for both the exponential decrease with depth and the exponential distribution horizontally.
The following references will be added: Beven & Kirkby (1979) and d'Orgeval et al. (2008) for the exponential decrease with depth (cf. C9); Entekhabi & Eagleson (1989) *and Vereecken et al., (2019)* for the horizontal distribution.

**C11:** - Line 70: please provide references for the evapotranspiration model.
The following references will be added: Krinner et al. (2005) for the modelling of evapotranspiration based on four sub-fluxes (L70-71); d'Orgeval et al. (2008), Campoy et al. (2013) to support the end of the paragraph explaining transpiration and soil evaporation are linked to soil moisture and properties.

**C12:** - Line 91: what is the error due to selecting only the dominant soil texture? Did the authors investigate the effect of upscaling by using some average (or weighted average) soil properties?
In this study, we did not aim at comparing different upscaling methods; it is out of the scope of this paper. However, in our discussion, we stated some studies which tested different upscaling methods (Samaniego et al., 2010; Montzka et al., 2017).

**C13:** - Line 113: "network owing to machine learning" – please rephrase.
We thank the reviewer for pointing out this mistake. It is corrected to: "network using machine learning"

Results:
**C14:** - Lines 133-135: the partitioning of rainfall in infiltration (soil moisture) and runoff differs among soil textures in a way that is well known and studied - I don't see the novelty here. Are the authors simply testing the model?
Yes, the first part of the results in the submitted paper was mostly intended to examine the response of the model to the different soil textures. This provides an additional evaluation of the recent version of ORCHIDEE with physically-based hydrology, which has not been heavily tested, as further explained in C19. However, an important outcome of this analysis is the non-monotonic response of evapotranspiration and total runoff to soil texture, since these two behaviors have not already been underlined, to our knowledge.

**C15:** - Lines 153-154: "Switching ... variables". If the maps are similar a priori, why would the authors expect any differences in the global water budget? It would probably be more useful, in my opinion, to focus on those areas where the maps are actually different and discuss the resulting biases in the hydrologic response in those areas.

The sentence following the cited one underlines that points with unchanged texture cover 41.2% of the land surface. It is a lot, but it leaves 58.8% where the soil texture does change from Reynolds to SoilGrids. That's why the weak sensitivity of the global water budget was not an expected result. Yet, we understand from the reviewers comments that the way we introduced the texture map similarities as an *a priori* explanation to the weak sensitivity of the simulated fluxes is misleading, and as suggested in C22, we will devote a new subsection at the beginning of the Results to gather quantified analyses of the similarities/differences between the tested texture maps.

Regarding the strong effect of soil texture in some local areas, it was already discussed in the conclusion of our submitted manuscript, but the suggestion of the reviewer to put a stronger emphasis on this kind of analysis is a good one. As already written (C1, C2, C4), we therefore propose to add a new subsection 3.1 in the Results to detail the effect of soil texture map change where the maps are different. A detailed description of this new sub-section is presented in C22.

**C16:** - Lines 170-177: the results discussed here could have been expected without running massive simulations: the partitioning of rainfall in infiltration and runoff with different soil textures is well known. The exercise here seems more of a reality check for the model than some novel analysis.

We agree that the partitioning of rainfall in infiltration and surface runoff with different soil textures is well known, and the global-scale averages discussed in the commented lines are indeed a reality check. We will shorten this discussion in the revised manuscript. The uniform experiments are more useful to analyze the importance of spatial variability of soil texture, as done in the paper based on Fig S2, to conclude that spatial patterns of simulated hydrologic variables are weakly driven by the soil texture, but rather by the climate forcing (L189-190). An important point, however, is that the largest difference in mean global scale ET between these uniform soil maps (between the uniform clay and silt experiments owing to the non-monotonic response underlined in Figs 3 and 4) is 0.1 mm/d, i.e 8% of the global mean ET using the complex soil texture maps and the same climate forcing. This tells us the maximum range of ET change we can expect from any kind of soil texture map change. This point was not stressed in the submitted manuscript, and will be added in the new version of the paper in sub-section 3.3.

**C17:** - In general, I think the paper lacks a proper quantification of the differences between the soil texture maps and the related bias in simulated fluxes. If the authors could provide a clear quantitative link between the bias in soil maps and the resulting bias in hydrologic partitioning this would actually allow to extrapolate something from the analysis. As it stands, the analysis only seems a modeling exercise without any useful application. I believe it would be more impactful if the paper could provide answers to questions like: how much does the hydrologic response (e.g., runoff, infiltration, etc) change if the soil texture differs by a certain percentage? How do the probability distributions of the water budget components vary with the distributions of soil texture?

The question is sound, but it is not easy to change soil texture by a certain percentage since it is a qualitative factor. We believe that Fig 4 of the submitted manuscript partially answers the reviewer's demand for quantification, in the special case of the switch from the Reynolds soil map

to SoilGrids.

[Figure]

*Figure A6 Maps of the standard deviation (SD) of (a) the logarithm of median particle diameter (dm) given by the three complex soil texture maps (Reynolds, Zobler, SoilGrids), and (b-h) the mean annual simulated variables (in mm/d except for soil moisture in mm) using the three different maps. The Spearman rank correlation between the SD of log(dm) and the SD of each variable is indicated above each map (Cor). The range of the color bar on each map extends from 0 to the rounded maximum value in the map.*

To go further, we propose to include a new result in section 3.3 "Spatial patterns of simulated fluxes and ET bias". The goal is to provide a point-scale quantification of the differences between the three complex soil maps on the one hand, and the resulting simulated variables on the other hand. To this end, we mapped the standard deviation (SD) of each group of three maps, using the mean diameter (*dm*) of each texture class to get a quantitative proxy in case of texture (Figure A6). Although the quantitative meaning of SD can be questioned when calculated from a sample of three values, we used it here as a lumped metric of similarity/difference between the three complex maps, and to identify points/regions where the three maps are all consistent (small SD), or where at least one of them is departing (high SD).

Figure A6 shows a resemblance between the map of soil texture SD and the SD maps of the simulated hydrologic variables, which was quantified at global scale with a spatial correlation

coefficient (ranging between 0.49 for transpiration to 0.79 for soil moisture). Areas which stand out in all SD maps are Central Africa, Central America, India and the Amazonian area, for which we propose a zoom in subsection 3.4 (as detailed in C4). Aside from these areas, the tropical humid zones show important differences in surface runoff and drainage (Figure A6f,h), but this is rather due to the high values of these fluxes in these very humid zones than to soil texture change.

**C18:** - The authors showed that hydrologic fluxes are more sensitive to changes in climatic forcings rather than soil texture maps. But how different are the climatic forcings used compared to soil texture maps? If the bias in the climatic forcing is, a priori, much higher, it is likely that the resulting hydrologic behavior will differ more.

We used two different climate forcings which are both widely used in the LSM community, and expected to be realistic. They both come from climate reanalysis including data assimilation, and are both bias corrected (L80-81). Both were selected for the off-line CMIP6 simulations (van den Hurk et al., 2016), which will added in the revised version (section 2.2). Thereby, the differences between these two climate datasets are not expected to be higher than ones between the three complex soil maps, also all intended to faithfully capture soil texture patterns. We could have used three state-of-the-art climate forcing datasets, but the resulting spread results would likely have been similar or larger (e.g. Guo et al.,2006; Yin et al., 2018, cited in the paper; Gelati et al, 2018). Thus, we believe it is a valid a conclusion that the uncertainty in the climate forcing exceeds the one of the soil texture maps (L189-190).

Conclusions:

**C19:** - Lines 197-198: the fact that the model has a realistic behavior should not be a main result. The orchidee model has been widely tested, and its ability to reproduce hydro-logic fluxes properly in relation to different soil textures is not a novel result.

We agree that the realistic behavior of the model is not the main result, but it serves to support the conclusions regarding the impact of soil texture maps in this model. We would also like to bring the reviewer's attention on the fact that the version of ORCHIDEE used in this study is based on a physical description of water fluxes using Richard's equation, which has not been as widely tested as the first version of ORCHIDEE based on a conceptual description of soil hydrology, especially given the fact that the model is always evolving. Besides, most papers which include an evaluation of the new soil physics from a hydrological perspective (de Rosnay et al., 2002; d'Orgeval et al., 2008; Boone et al., 2009, Campoy et al., 2013; Guimberteau et al., 2014; Barella-Ortiz et al., 2017; Raoult et al., 2018) did not focus on the rightful sensitivity to soil texture.

**C20:** - Lines 210-212: What is the point of using spatially similar maps to see if they have any discernible effects on the hydrologic fluxes? If, a priori, the maps are similar, what is the point of the entire exercise?

This question is answered earlier in C16.

**C21:** - Line 214: Did the authors try to test some weighted average SHPs thus accounting for spatial variability instead of using the dominant soil texture in each cell?

This question is answered earlier in C11.

**C22:** - Line 225: A detailed analysis of the difference between the various maps should be given upfront. This only appears with Fig. 8 but it would be beneficial to have an in depth analysis of key differences among these maps (as well as of differences resulting from adopting different

strategies for upscaling the higher resolution maps) at the beginning of the manuscript.

As suggested by the reviewer, we propose to add, in the revised version, a new sub-section 3.1 in the Results called *"Comparison of the tested soil texture maps", and* dedicated to a quantified analysis of the differences and similarities of the tested texture maps. This sub-section will discuss Figure 8 of the submitted manuscript, which will be updated as follows: Fig. 8a will become Table 3 but remain unchanged (percent overlap between the tested maps), and Fig.8b (to become Table 4; following Table A1 below) will be expanded to show the spatial correlation coefficients between the maps of not only *dm*, but also of other important hydraulic parameters deduced from the soil texture maps: $K_s$ (as provided by the PTF, thus not including the impact of roots nor soil compaction), soil porosity $\theta_s$, field capacity $\theta_{fc}$, wilting point $\theta_w$, and available water content (AWC, integrated over the the 2m soil column).

Table A1 Mean and SD of soil parameter maps corresponding to the three different soil texture maps; and correlation coefficients between these maps of soil parameters.

| | | SoilGrids | Reynolds | Zobler |
|---|---|---|---|---|
| **log(*dm*)** | Mean (log μm) | 4.3 | 4.3 | 4.2 |
| | SD (log μm) | 1.39 | 1.6 | 1.1 |
| | Cor. SoilGrids | 1.00 | 0.27 | 0.34 |
| | Cor. Reynolds | 0.27 | 1.00 | 0.43 |
| **$K_s$** | Mean (mm/d) | 628 | 708 | 417 |
| | SD (mm/d) | 1355 | 1263 | 361 |
| | Cor. SoilGrids | 1.00 | 0.30 | 0.35 |
| | Cor. Reynolds | 0.30 | 1.00 | 0.34 |
| **Saturated water content** | Mean ($m^3/m^3$) | 0.419 | 0.418 | 0.423 |
| | SD ($m^3/m^3$) | 0.018 | 0.017 | 0.010 |
| | Cor. SoilGrids | 1.00 | 0.35 | 0.24 |
| | Cor. Reynolds | 0.35 | 1.0 | 0.31 |
| **Field capacity** | Mean ($m^3/m^3$) | 0.177 | 0.176 | 0.168 |
| | SD ($m^3/m^3$) | 0.059 | 0.067 | 0.042 |
| | Cor. SoilGrids | 1.00 | 0.31 | 0.35 |
| | Cor. Reynolds | 0.31 | 1.00 | 0.57 |
| **Wilting point** | Mean ($m^3/m^3$) | 0.100 | 0.102 | 0.091 |
| | SD ($m^3/m^3$) | 0.039 | 0.049 | 0.024 |
| | Cor. SoilGrids | 1.00 | 0.40 | 0.34 |
| | Cor. Reynolds | 0.40 | 1.00 | 0.59 |
| **AWC** | Mean (mm) | 154.7 | 147.8 | 154.8 |
| | SD (mm) | 60.9 | 57.4 | 36.4 |
| | Cor. SoilGrids | 1.00 | 0.06 | 0.24 |
| | Cor. Reynolds | 0.06 | 1.00 | 0.34 |

This table shows that, whichever the soil parameter, the difference of spatial mean between the three maps is smaller than the difference of spatial standard deviation (SD), even for the least variable map (Zobler). This demonstrates the large similarity of the three complex soil maps tested in our paper, not only regarding soil texture itself (as summarized by *dm*), but also, very logically, for the derived soil hydraulic parameters. This similarity is also confirmed by the spatial correlations between each pair of maps, always positive, the best correlations being found between Zobler and Reynolds (always larger than 0.3, and up to 0.59), and the weakest between SoilGrids and Reynolds. Among the different soil parameters, the AWC shows the lowest spatial correlations, which probably comes from the fact that the maximum AWCs are found for medium textures (Fig. S1), which are the dominant textures in all the three maps (Table 1). The above-mentioned results as well as Table A1 will be added in the new section 3.1 of the revised manuscript.

**C23:** - The clay bias that is only briefly discussed in lines 225-230 seems actually a quite interesting point.

We agree that clay related features were only briefly discussed in the submitted manuscript. In the revised version, we will add a sub-section where we zoom on those areas where clay has a non-negligible impact (details in C4).

**C24:** If the prevalence of loamy texture in the texture maps is – in part – an artifact due to upscaling procedures and averaging, what would the bias be in the hydrologic partitioning if the actual texture in some grid cell was not as loamy as assumed?

Based on Figures 3 and 4 of the submitted manuscript, we have shown that ET and total runoff show a non-monotonic behavior with respect to soil texture classes (sorted by increasing median diameter); the latter result is a major finding of our manuscript, as highlighted in the abstract. Since the loamy textures correspond to a lower total runoff and higher ET, an overestimation of the loamy textures in a soil texture map would lead to a systematic positive bias in ET and negative bias in total runoff.

**C25:** - Lines 239-240: Some products (e.g., SoilGrids) have vertically variable information on soil texture – why didn't the authors use this information to relax the hypothesis of vertically homogeneous texture?

We answered this question in C9.

**C26:** - Lines 236 – 244: most of the paper focused on soil texture, while only two PTFs were tested. Why is the conclusive paragraph of the manuscript on PTFs and inclusion of additional factors in currently used PTFs, while the manuscript only slightly touched this point? Although this is an interesting topic, I wouldn't embark into a discussion on PTFs at this point of the manuscript (as the authors didn't actually do an in depth analysis of the bias induced by different PTFs).

It is true that we only considered two different PTFs in our simulations (EXP4 in and EXP5). The reason is technical, because of the SP-MIP protocol, but we did not focus our analysis on the resulting changes, which will be explored within the SP-MIP project. In this framework, our main conclusions address the impact of the soil texture maps on the simulated land hydrology, and the goal of the last paragraph of the paper is to open the discussion to the impact of other soil properties, such as bulk density and soil structure. Thus, using other sources of information than soil texture to derive the geographic distribution of soil properties may lead to clearer and broader improvements of the simulated water budget than the ones analyzed here owing to soil texture maps alone. We propose to replace the last sentence of the paper by the above one.

**References:**

Barella-Ortiz, A., Polcher, J., Rosnay, P. de, Piles, M. and Gelati, E.: Comparison of measured brightness temperatures from SMOS with modelled ones from ORCHIDEE and H-TESSEL over the Iberian Peninsula, Hydrol. Earth Syst. Sci., 21(1), 357–375, doi:https://doi.org/10.5194/hess-21-357-2017, 2017.

Beven, K. J. and Kirkby, M. J.: A physically based, variable contributing area model of basin hydrology / Un modèle à base physique de zone d'appel variable de l'hydrologie du bassin versant, Hydrol. Sci. Bull., 24(1), 43–69, doi:10.1080/02626667909491834, 1979.

Bierkens, M. F. P., Bell, V. A., Burek, P., Chaney, N., Condon, L. E., David, C. H., Roo, A. de, Döll, P., Drost, N., Famiglietti, J. S., Flörke, M., Gochis, D. J., Houser, P., Hut, R., Keune, J., Kollet, S., Maxwell, R. M., Reager, J. T., Samaniego, L., Sudicky, E., Sutanudjaja, E. H., Giesen, N. van de, Winsemius, H. and Wood, E. F.: Hyper-resolution global hydrological modelling: what is next?, Hydrol. Process., 29(2), 310–320, doi:10.1002/hyp.10391, 2015.

Campoy, A., Ducharne, A., Cheruy, F., Hourdin, F., Polcher, J. and Dupont, J. C.: Response of land surface fluxes and precipitation to different soil bottom hydrological conditions in a general circulation model, J. Geophys. Res. Atmospheres, 118(19), 10,725-10,739, doi:10.1002/jgrd.50627, 2013.

De Lannoy, G. J. M., Koster, R. D., Reichle, R. H., Mahanama, S. P. P. and Liu, Q.: An updated treatment of soil texture and associated hydraulic properties in a global land modeling system, J. Adv. Model. Earth Syst., 6(4), 957–979, doi:10.1002/2014MS000330, 2014.

D'Orgeval, T., Polcher, J. and Rosnay, P. de: Sensitivity of the West African hydrological cycle in ORCHIDEE to infiltration processes, Hydrol. Earth Syst. Sci., 12(6), 1387–1401, doi:https://doi.org/10.5194/hess-12-1387-2008, 2008.

Ducharne, A., Ghattas, J., Maignan, F., Ottlé, C., Vuichard, N., Guimberteau, M., Krinner, G., Polcher, J., Tafasca, S., Bastrikov, V., Brender, P., Cheruy, F., Guénet, B., Mizuochi, H., Peylin, P., Tootchi, A. and Wang, F.: Soil water processes in the ORCHIDEE-2.0 land surface model: state of the art for CMIP6, Geosci. Model Dev., in prep.

Entekhabi, D. and Eagleson, P. S.: Land Surface Hydrology Parameterization for Atmospheric General Circulation models Including Subgrid Scale Spatial Variability, J. Clim., 2(8), 816–831, doi:10.1175/1520-0442(1989)002<0816:LSHPFA>2.0.CO;2, 1989.

Gelati, E., Decharme, B., Calvet, J.-C., Minvielle, M., Polcher, J., Fairbairn, D. and Weedon, G. P.: Hydrological assessment of atmospheric forcing uncertainty in the Euro-Mediterranean area using a land surface model, Hydrol. Earth Syst. Sci., 22(4), 2091–2115, doi:10.5194/hess-22-2091-2018, 2018.

Guo, Z., Dirmeyer, P. A., Hu, Z.-Z., Gao, X. and Zhao, M.: Evaluation of the Second Global Soil Wetness Project soil moisture simulations: 2. Sensitivity to external meteorological forcing, J. Geophys. Res. Atmospheres, 111(D22), doi:10.1029/2006JD007845, 2006.

Hengl, T., Jesus, J. M. de, Heuvelink, G. B. M., Gonzalez, M. R., Kilibarda, M., Blagotić, A., Shangguan, W., Wright, M. N., Geng, X., Bauer-Marschallinger, B., Guevara, M. A., Vargas, R., MacMillan, R. A., Batjes, N. H., Leenaars, J. G. B., Ribeiro, E., Wheeler, I., Mantel, S. and Kempen, B.: SoilGrids250m: Global gridded soil information based on machine learning, PLOS ONE, 12(2), e0169748, doi:10.1371/journal.pone.0169748, 2017.

Kirkby, M. J. and Beven, K. J.: A physically based, variable contributing area model of basin hydrology, Hydrol. Sci. J., 24, 43–69, 1979.

Krinner, G., Viovy, N., Noblet-Ducoudré, N. de, Ogée, J., Polcher, J., Friedlingstein, P., Ciais, P., Sitch, S. and Prentice, I. C.: A dynamic global vegetation model for studies of the coupled atmosphere-biosphere system, Glob. Biogeochem. Cycles, 19(1), doi:10.1029/2003GB002199, 2005.

Montzka, C., Herbst, M., Weihermüller, L., Verhoef, A. and Vereecken, H.: A global data set of soil hydraulic properties and sub-grid variability of soil water retention and hydraulic conductivity curves, Earth Syst. Sci. Data, 9(2), 529–543, doi:https://doi.org/10.5194/essd-9-529-2017, 2017.

Raoult, N., Delorme, B., Ottlé, C., Peylin, P., Bastrikov, V., Maugis, P. and Polcher, J.: Confronting Soil Moisture Dynamics from the ORCHIDEE Land Surface Model With the ESA-CCI Product: Perspectives for Data Assimilation, Remote Sens., 10(11), 1786, doi:10.3390/rs10111786, 2018.

Samaniego, L., Kumar, R. and Attinger, S.: Multiscale parameter regionalization of a grid-based hydrologic model at the mesoscale, Water Resour. Res., 46(5), doi:10.1029/2008WR007327, 2010.

Van den Hurk, B., Kim, H., Krinner, G., Seneviratne, S. I., Derksen, C., Oki, T., Douville, H., Colin, J., Ducharne, A., Cheruy, F., Viovy, N., Puma, M. J., Wada, Y., Li, W., Jia, B., Alessandri, A., Lawrence, D. M., Weedon, G. P., Ellis, R., Hagemann, S., Mao, J., Flanner, M. G., Zampieri, M., Materia, S., Law, R. M. and Sheffield, J.: LS3MIP (v1.0) contribution to CMIP6: the Land Surface, Snow and Soil moisture Model Intercomparison Project – aims, setup and expected outcome, Geosci Model Dev, 9(8), 2809–2832, doi:10.5194/gmd-9-2809-2016, 2016.

Vereecken, H., Weihermüller, L., Assouline, S., Šimůnek, J., Verhoef, A., Herbst, M., Archer, N., Mohanty, B., Montzka, C., Vanderborght, J., Balsamo, G., Bechtold, M., Boone, A., Chadburn, S., Cuntz, M., Decharme, B., Ducharne, A., Ek, M., Garrigues, S., Goergen, K., Ingwersen, J., Kollet, S., Lawrence, D. M., Li, Q., Or, D., Swenson, S., de Vrese, P., Walko, R., Wu, Y. and Xue, Y.: Infiltration from the Pedon to Global Grid Scales: An Overview and Outlook for Land Surface Modeling, Vadose Zone J., 18(1), doi:10.2136/vzj2018.10.0191, 2019.

Wood, E. F., Roundy, J. K., Troy, T. J., Beek, L. P. H. van, Bierkens, M. F. P., Blyth, E., Roo, A. de, Döll, P., Ek, M., Famiglietti, J., Gochis, D., Giesen, N. van de, Houser, P., Jaffé, P. R., Kollet, S., Lehner, B., Lettenmaier, D. P., Peters-Lidard, C., Sivapalan, M., Sheffield, J., Wade, A. and Whitehead, P.: Hyperresolution global land surface modeling: Meeting a grand challenge for monitoring Earth's terrestrial water, Water Resour. Res., 47(5), doi:10.1029/2010WR010090, 2011.

Yin, Z., Ottle, C., Ciais, P., Guimberteau, M., Wang, X., Zhu, D., Maignan, F., Peng, S., Piao, S., Polcher, J., Zhou, F. and Kim, H.: Evaluation of ORCHIDEE-MICT-simulated soil moisture over China and impacts of different atmospheric forcing data, Hydrol. Earth Syst. Sci., 22(10), 5463–5484, doi:10.5194/hess-22-5463-2018, 2018.

---

## Author Comment (AC3) · 20 Jan 2020

**Weak sensitivity of the terrestrial water budget to global soil texture maps in the ORCHIDEE land surface model**

Salma TAFASCA[1] , Agnès DUCHARNE[1] , Christian VALENTIN[2]

*Correspondence to:* Salma Tafasca (salma.tafasca@upmc.fr)

**Reply to anonymous Referee #3**

**General**

**C1:** This paper explores the impact of soil texture on the simulated water budget by the ORCHIDEE LSM at the global scale at 0.5 degree resolution. The authors conclude that the use of three different soil texture maps result in very similar terrestrial water budgets, and that the choice of the input soil texture map is not crucial for large scale modelling. While the study topic is very relevant and deserves publication, the manuscript needs to be revised.

We would like to thank the reviewer for taking the time to go through the paper, and for the relevant comments. According to all the three reviews that we received, we decided to make some substantial changes to the paper, in particular, the scientific question of the paper will be more clarified and new sub-sections will be added. A detailed presentation of the new structure of the paper is presented in the answer to Referee #1. In the following, we will provide a response to every point raised by Referee #3, these points are numbered from C1 to C21 for convenience.

**C2:** First, I think the authors should make in the Introduction their research question(s) clearer, in my opinion lines 52-53 are not sufficient, and it is also not quite clear why this research is different from earlier studies.

This comment was already raised by Referee #1, and we agree that the scientific question of the paper was too briefly stated in the introduction of the submitted paper. In the revised version of our paper, this will be clarified by expanding the last line of the introduction to a more classical paragraph detailing the specific research question of the paper and the structure of the paper: "*Here, we aim at exploring more systematically the impact of soil texture on the water budget from point to global scale, using a state-of-the-art LSM with physically-based soil hydrology, and multiple input soil texture maps. After presenting the model and soil texture maps used in this work, the results are presented as follows. We first provide an analysis of the similarities and differences between the different soil maps, then, we evaluate the point-scale response of the model to different soil textures to make sure it displays a reliable behavior. This point-scale response is then analyzed from a geographic point of view, with a comparison to a distributed observation-based ET product, and a focus is made on areas with a large sensitivity to the soil texture maps. We finally explore how the magnitude and significance of the simulated ET changes with the scale of analysis up to the land scale, defining the terrestrial water budget. The closing section summarizes the main conclusions of the study, and discusses its limitations and perspectives.*"

**C3:** The authors mention the use a physically based soil hydrological modelling component (including Richards equation) in these lines (line 53), but do not follow up in the Discussion and conclusions section.

This is a good point, and we propose to add the following sentences in the closing section, at the end of L210, after discussing the SP-MIP project: *"As mentioned in Introduction, much stronger responses to soil properties have been reported from bucket-type LSMs. It must be underlined, however, that these papers considered much larger changes of soil properties, which reduces in bucket-type models to available water holding capacity (AWC), combining information on porosity, soil depth, and the difference between field capacity and wilting point. As an example, the main changes discussed in Stamm et al. (1994), Ducharne & Laval (2000), de Rosnay & Polcher (1998), and Milly & Dunne (1994), correspond respectively to AWC changes of +75%, +110%, +200%, and +1400%, while the AWC changes when switching among the three soil texture maps used in the present paper range between +1 and +7%."* These percentages will be supported by citing the updated Figure 8, to be moved to the new section 3.1, cf response to comment C22 Referee#1.

**C4:** Furthermore, I believe the manuscript could benefit from a more detailed analysis and description on the differences between hydrological variables from different soil texture inputs (and PTFs), also focused on a regional/local scale.

We agree with the reviewer, and as detailed in the answer to Referee #1, we will add a dedicated sub-section in the results (3.4) where we look into the impacts of soil texture in the most impacted regions.

**C5:** Finally, I am wondering why the authors did choose to scale up the high resolution soil texture dataset to the model resolution as a function of the dominant USDA soil texture class. Why not, when applicable, calculate the soil hydraulic parameters at high resolution, and then scale up (with appropriate scaling operators (for example in the line with Samaniego et al., 2010))?

By default, ORCHIDEE upscales the input soil texture map to the model resolution (0.5°) by selecting the dominant USDA soil texture. This choice is hard-coded, and it is not in the purpose of our study to test different upscaling methods. However, we would like to point out that only the Reynolds map was upscaled by the model; the Zobler soil map is available at 1° resolution, so no upscaling was performed for this map, and the used SoilGrids map was provided by the SP-MIP team at the 0.5° resolution.

**Specific comments**

**C6:** A detailed description of the ORCHIDEE LSM would be helpful.

Based on this comment, and the ones from Referee #2 regarding root uptake and the effect of LAI on evapotranspiration, we will expand the description of ORCHIDEE in section 2.1. To this end, lines 61-74 will be changed to the following text (changes in bold):

[revised manuscript text omitted]

**C7:** Line 13: explain "medium texture", I think not every reader knows what medium means
Medium textures are the loamy textures, with medium *dm* (median diameter). To clarify this in the abstract, we will use the term loamy texture, which is clearer.

**C8:** Line 13-14: "The three tested complex soil texture maps being rather similar by construction…". Do the authors mean that the soil texture maps are similar because of the way how they were constructed (taking the dominant USDA soil texture class)?

The soil texture maps are similar because of the way they are upscaled but also, and more importantly, because of their common origins (FAO/UNESCO soil map). Based on Referee #1 comments, we think that these lines are misleading, since they suggest that the similarity between the soil texture maps is *a priori* knowledge of this study, while it is a result. These lines will be replaced by a new sub-section, inserted in the beginning of the Results, and gathering quantified analyses of the similarities/differences between the tested texture maps (cf. answer to Referee #1).

**C9:** As mentioned in the General comments, why not calculate soil hydraulic parameters at the high resolution scale?

As mentioned earlier, testing different upscaling methods is out of the scope of this paper.

**C10:** Indeed the soil texture maps are quite similar. Why then not focus more on sensitivity of PTFs (now two are used in this study)?

The sensitivity to various PTFs has been the scope of many studies, as recently reviewed by van Looy et al. (2017). In contrast, the main objective of our study is to examine the hydrological response to different soil texture maps. We consider two different PTFs in our simulations (EXP4 and EXP5) because of the SP-MIP protocol, but we don't focus our analysis on the resulting changes, which will be explored within the SP-MIP project, and are very weak based on land averages, but for soil moisture, noted by the Referee in comment C15.

**C11:** Line 35: 1-km SoilGrids database. A 250 m version is also available. Were the different soil layers also included in the analysis? And if yes, how? Also for example to calculate the exponential decline of Ks?

We will mention the availability of the 250m version of SoilGrids (Hengl et al., 2017) at line 35. As said in the paper, the SoilGrids map used in this study was processed at 0.5° for the SP-MIP project, and we will add it is based on the texture at 0cm depth (section 2.2). But even if SP-MIP had provided soil textures for different horizons, this information cannot be used in ORCHIDEE, as explained in the description of the model, in the revised version of the paper (cf. response to C6): "*Soil texture is assumed to be uniform over the soil column in ORCHIDEE, which does not permit to distinguish several soil horizons. However, $K_s$ decreases exponentially with depth, to account for the effects of soil compaction and bioturbation, as introduced by d'Orgeval et al. (2008) following Beven & Kirkby (1979).*" We also underline that the simplifying hypothesis of a uniform texture over the whole soil column is discussed in the concluding section of the submitted manuscript (lines 239-240).

**C12:** Lines 144-145: "Rather surprisingly, we find here…", Could you explain this in more detail? If drainage and transpiration decrease you would expect higher soil moisture values, right? The transpiration decrease is perhaps controlled by dominant vegetation type?

We agree with the reviewer, and will thus remove "Rather surprisingly". As for the transpiration decrease for fine-textured soils, it is not controlled by dominant vegetation types, as supported by Figure 4c where each pixel of the matrix corresponds to a unique set of grid-points undergoing a soil texture change, thus with unchanged climate and vegetation

cover. This implies that the decrease of transpiration found in Figure 3 when soil texture gets finer is effectively due to soil texture. A likely reason is the increase of matric potential, thus soil moisture retention, when the texture gets finer, as shown in Figure S1 for particular values of the potential, defining the wilting point, field capacity and air entry suction point (1/α). This analysis leads to a more complex explanation of the response of transpiration to soil texture, and we propose to replace lines 142-147 by the following paragraph:
*"Transpiration, however, increases as soil gets coarser (Fig. 3c), with two explanations probably acting together. Firstly, the increase of matric potential when the texture gets finer, as shown in Figure S1 for particular values of the potential, defining the wilting point, field capacity and air entry suction point (1/α), makes root uptake thus transpiration more difficult for a given soil moisture if the soil texture is finer. Secondly, the high conductivity of coarse soils enhances water infiltration at the soil surface, quickly available for plant uptake. The increase of Ks for coarse textures also explains the associated drainage increase when its dependence on mean precipitation is filtered (Fig. 3f). The fact that soil moisture decreases when drainage and transpiration get higher indicates that annual mean soil moisture is the result more than the cause of these fluxes"*.

**C13:** Lines 148-149: Could you elaborate more (also include references)? These factors should also affect evapotranspiration…
To support this sentence, focused on the response of soil evaporation, we propose to add the following references: Martens et al. (2017) and Wang et al (2018) regarding the anti-correlation between LAI and soil evaporation (further supported by the spatial correlation of -0.32 between these two variables in our simulation EXP2); the negative impact of vegetation on soil evaporation can also develop owing to the litter, which exerts a resistance to this flux (Ogée & Brunet, 2002; Sakaguchi & Zeng, 2009). However, the dependence of soil evaporation on climatic variable (temperature, potential ET) and soil moisture will not be expanded, as it is very well established. Then, the referee is right that this dispersion transfers to evapotranspiration (Figure 3d), but to a weaker extent since soil evaporation is not the main component of total ET.

**C14:** Line 158: Please describe Figure 4 in more detail.
We agree with Referee #3 (and Referee #2) that Figure 4 was too briefly discussed. This figure is intended to show how the simulated variables change when only soil texture changes, to better analyze the model's response to the different soil textures. We propose to expand the last paragraph of section 3.1 addressing this Figure:
*"By focusing this time on the point-scale changes induced by changing the soil texture map (from Reynolds to SoilGrids), Figure 4 highlights that the simulated soil evaporation decreases from fine to coarse textures, so that capillary retention, which is the main limiting factor to soil evaporation in ORCHIDEE, depends more strongly on soil moisture (higher for fine soils) than on intrinsic capillary forces (stronger for fine soils). We fail to see this behavior in Figure 3, which is likely due to the greater impact of diverse climatic conditions and vegetation associated with every soil texture. Figure 4 also confirms the results of Figure 3 for the other variables, including the decrease of soil moisture with coarser soils and the greater impact of soil texture on runoff variables (surface runoff and drainage).  In particular, we find that replacing fine textures with coarse textures (above the first diagonal of the matrices) results in higher drainage (due to the higher permeability of coarse-textured soils) and lower surface runoff, with changes that can exceed 1mm/d in absolute value for some textural*

*changes (all involving medium texture classes). As a result, less water is available in the soil, which leads to less soil evaporation, further leading to more transpiration (Fig. 4bc).*

*The convex behavior of total runoff with soil texture can also be seen in Figure 4h, which is antisymmetric along the two diagonals, thus defining four different kinds of total runoff change to soil texture change. This behavior results from the fact that total runoff sums up two variables of opposite response to soil texture change (surface runoff and drainage), the net response depending on the dominant component. Hence, changes to medium textures from either coarse or fine textures (left and right red triangles in Fig. 4h) lead to reduced total runoff, owing to reduced surface runoff in the first case, and reduced drainage in the second. In contrast, changes from medium texture to either coarse or fine textures lead to increased runoff (bottom and top blue triangles in Fig. 4h), owing to increased surface runoff or drainage, respectively. This pattern thus means that the medium textures correspond the smallest total runoff. By means of long-term water conservation, the opposite patterns are found for total evapotranspiration changes (Figure 4d), because of the opposite responses of soil evaporation and transpiration to soil texture, and supporting the concave response of this flux to soil texture found in Figure 3.*"

**C15:** Figure 5: EXP5 seems to show a large difference in soil moisture with EXP4. In my opinion an interesting result, but not mentioned in the text and explained.

While both EXP4 and EXP5 use the same soil texture map, the PTF used in each experiment is different. Moreover, in EXP5 (as well as EXP6-EXP9), Ks is constant with depth, unlike the experiments EXP1-EXP4. We will add this clarification when describing the different simulations (section 2.2, L 1109): "*It must be noted that the five simulations based on the soil parameters of Schaap et al. (2001) also differ from the four others (EXP1 to EXP4) because the decrease of Ks with depth is relaxed, to comply with the SP-MIP protocol.*"

As a consequence, the decrease of soil moisture between EXP4 and EXP5 is not only due to PTF change but also the increase of Ks at the bottom of the soil column in EXP5, because Ks does not decrease with depth. This favors drainage, thus reduces soil moisture, and we propose to add this explanation when discussing Fig. 5, at line 172.

**C16:** Line 187-188 and Figure 7: Ok, indeed transpiration and soil evaporation show weak sensitivity, but other variables like drainage, surface runoff and soil moisture show a stronger sensitivity. For example, when you focus on Scandinavia, drainage decreases, surface runoff increases, and soil moisture increases. I believe the manuscript should also focus on these variables, in specific regions. Why is transpiration not affected here by the soil texture maps, and the water balance components as drainage, surface runoff and soil moisture do change?

As stated earlier, a new subsection dedicated to regional zooms on the most impacted regions by soil texture change will be added in the Results (3.4). In Figure 7, the Scandinavian soils were changed from Sandy Loam (in Reynolds map) to Loam (in Zobler map). According to Figure 4, the consequence of this change is an increase in surface runoff (by 0.1-0.5 mm/d) and soil moisture (by 100-200 kg/m$^2$), and a decrease in drainage (by -0.1 to -0.05 mm/d). The change in transpiration is much lower than the one of surface runoff and drainage, and does not exceed 0.05 mm/d (in absolute value). The latter results are well in agreement with Figure 7, and the non-significant changes in transpiration pointed by the Referee are in fact due to the weak impact of soil texture change on transpiration.

**C17:** Lines 208-209: what about other variables (water balance components) than evapotranspiration?
Up to now, only preliminary results were communicated by the SP-MIP team. No information about other variables was revealed.

**C18:** Lines 214-215: why not calculate hydraulic parameters at high resolution to remove some of that bias?
As explained earlier, it is out of the purpose of the paper to test different upscaling methods.

**C19:** Lines 218-219: yes, the authors could have used these upscaling method of Samaniego.
In this paper, we do not aim at testing different upscaling methods.

**C20:** Lines 226-235: again the focus on evapotranspiration. What about other water balance components?
In our paper, we mapped the impact of soil texture on different hydrologic variables (Figure 7 of the submitted paper and Figure S3 of the supplementary), but we chose to map the biases of the ET variable since the distributed observation-based products are only available for this variable.

**C21:** Line 236-244: To include and end with this paragraph the authors should focus more on PTFs (methods and results) and describe these in more detail (methods).
It must be underlined that we don't claim here that our paper demonstrates the need for more complex PTFs. On the contrary, it massively cites other studies supporting this conclusion, and we do so because the need for more complex PTFs is related to the specific conclusion of our paper, i.e. the weak sensitivity to soil texture maps except in some very specific areas where the USDA class for Clay is not precise enough. Thus, using other sources of information than soil texture to derive the geographic distribution of soil properties may lead to clearer and broader improvements of the simulated water budget than the ones analyzed here owing to soil texture maps alone. We propose to replace the last sentence of the paper by the above one.